

# Integrating aquatic and terrestrial biogeochemical model to predict effects of reservoir creation on CO₂ emissions

Weifeng Wang[1], Nigel T. Roulet[1,2], Youngil Kim[3], Ian B. Strachan[4], Paul del Giorgio[5], Yves T. Prairie[5], Alain Tremblay[6]

[1] Department of Geography, McGill University, Montréal, QC, H3A 0B9 Canada
[2] Centre d'Études Nordiques, Université Laval, Québec, QC, G1V 0A6, Canada
[3] Department of Forest Ecosystems and Society, Oregon State University, Corvallis, OR, 97330, USA
[4] Department of Natural Resource Sciences, McGill University, Ste Anne de Bellevue, Montréal, QC H9X 3V9, Canada
[5] Département des Sciences Biologiques, Université du Québec à Montréal, Case Postale 8888, Succ Centre-Ville, Montréal, QC H3C 3P8, Canada
[6] Environment Production, Hydro-Québec, Montreal, QC H2Z 1A4, Canada

*Correspondence to*: Weifeng Wang (weifeng.wang@mcgill.ca)

**Abstract.** There is considerable debate on the role of hydroelectric reservoirs for the emission of CO₂ and other greenhouse gases. To quantify CO₂ emissions from a newly created reservoir that was formed by flooding the boreal landscape we developed a daily time-step reservoir model by integrating a terrestrial and an aquatic ecosystem model. We calibrated the model using the measurements of dissolved organic and inorganic carbon (C) in a ~600 km² boreal hydroelectric reservoir, Eastmain-1, in northern Quebec, Canada. A major constraint we dealt with is the dearth of basic environmental data for the Boreal region so we took a parsimonious approach for required inputs. We then evaluated the model performance against observed CO₂ fluxes data from an eddy covariance tower in the middle of the EM-1 reservoir for the period from 2006 to 2012 and compared internal variables such as water column respiration, chlorophyll-*a* concentration, and sedimentation rate to measurements from field campaigns during 2006–2008. The model predicted the seasonal and inter-annual variability of CO₂ emissions reasonably well compared to the observations. Discrepancies between simulation results and observations usually occurred near ice-off dates when there was large amount of dissolved CO₂ under ice-cover. We applied the model to assess the effects of reservoir creation on C dynamics over the estimated "engineering" reservoir lifetime (i.e., 100 years). We found that the reservoir acts as a net C source over its lifetime and simulated CO₂ fluxes were 204 g C m⁻² yr⁻¹ in the first year after flooding, steeply declined in the first three years, and then steadily decreased to ~110 g C m⁻² yr⁻¹ with increasing reservoir age. Sensitivity analyses revealed that the amount of terrestrial organic C flooded and oxygen effects can positively enhance benthic respiration and CO₂ fluxes across air–water interface, but the effects on CO₂ emissions were not significant. Higher temperatures dramatically stimulate CO₂ emissions by enhancing CO₂ production in both the water column and the sediment, and extending the duration of the open water period over which emissions can occur. Changing wind speeds had large uncertainties on annual CO₂ emissions, given that wind speeds not only affect the gas transfer rate but also the open water period by affecting the surface energy balance. The model is useful for the estimation of CO₂ emissions from reservoirs to the





atmosphere and could be used to assist the hydro-power industry and others interested in emissions evaluate the role of boreal reservoirs as sources of greenhouse gas emissions.

## 1 Introduction

Water reservoirs, especially hydroelectric reservoirs, have become a focus of attention because these artificial
lakes that were formed by flooding land (e.g., forests and wetlands) emit greenhouse gases (GHGs) (Barros et al., 2011;St. Louis et al., 2000). Terrestrial ecosystems usually take up carbon (C) dioxide to amass a significant store of C in living biomass, litters, and soils, but rapidly release a portion of the store when disturbed (e.g. shifts in climate, fire, insects, blowdown, ice storms, etc.) (Kurz et al., 2013;Bradshaw and Warkentin, 2015), while aquatic ecosystems (e.g., lakes) often emit C (Algesten et al., 2004;Wik et al., 2016) receiving much of their C from the surrounding
catchment. Moreover, flooding terrestrial ecosystems eliminates terrestrial uptake of C and converts stores of terrestrial C to water-saturated sediments where the organic matters (e.g., plant biomass, litter, and soil organic matter) decompose and then emit to the atmosphere (Brothers et al., 2012b). Such land-use change can rapidly alter and create novel environmental conditions that fundamentally alter the carbon cycle (Teodoru et al., 2012). Since hydro-electricity is proposed as a viable non-fossil fuel based source of energy for the future, managers and policy-makers require GHG
emission assessments from reservoirs (Liden, 2013) and scientific communities need to understand how the land-use change alters the C cycle.

Many studies of greenhouse gas fluxes from reservoirs have demonstrated that most reservoirs are the C source to the atmosphere (e.g., Barros et al., 2011). It has been suggested that young reservoirs emit higher GHGs than old ones (Barros et al., 2011;St. Louis et al., 2000), because flooded terrestrial organic matters not only contribute large
portion of dissolved $CO_2$ but subsequently provide the water column dissolved organic carbon (DOC) that enhances water column respiration in young reservoirs (Brothers et al., 2012b). However, the effects of flooding terrestrial ecosystems to create a reservoir on C processing in the flooded soil and the water column are still poorly understood and the change of $CO_2$ emissions with reservoir age is highly uncertain (Barros et al., 2011;Kim et al., 2016). Given that reservoirs have a lifetime of up to 100 years and they will be subjected, it is important to develop a mechanistic-
based model that is capable of simulating the $CO_2$ exchange across their life-time considering the land-use change from terrestrial ecosystems to a reservoir.

The environmental factors such as temperatures and wind speeds can influence the processing and transport of C in inland waters and therefore play an important role in regulating gas exchange across air−water interface (Åberg et al., 2010;Greene et al., 2014). For example, ice cover impedes methane ebullition bubbles in a seasonally ice-covered
lakes (Greene et al., 2014); water turnover could transport dissolved $CO_2$ from the deeper to upper water (Eugster et al., 2003), resulting in peak emissions in late summer and autumn for a boreal lake (Huotari et al., 2011). However, the seasonal variability of $CO_2$ emissions from boreal reservoirs is still unclear and depends on interactions between



biogeochemical and physical processes that we still do not understand in these newly created ecosystems under climate change conditions (e.g., increasing temperature). Therefore, estimates of seasonal gas exchange require a mechanical model that is able to simulate both processes governing the C processing and transport in these managed aquatic ecosystems.

5       In this study, a process-based model that can simulate both physical and biogeochemical processes for reservoirs is developed. The model we present, the *F*orest *Aq*uatic-*De*nitrification *Dec*omposition model (FAQ-DNDC), adapts and adds to a well-known terrestrial ecosystem model (Li et al., 1992) for the conditions found when terrestrial ecosystems are flooded. Eventually, we wish to simulate the possible long-term (~century) net $CO_2$ emissions (i.e., differences between post- and pre-impoundment balance of $CO_2$ emissions) from northern boreal landscapes following

procedures and protocols to quantitatively analyze net GHG emissions (International Energy Agency, 2015). We thus developed a new model rather than using existing reservoir models such as CE-QUAL-W2 (Cole and Wells, 2006) and Delft3D-ECO (Los et al., 2008) because we want the model that uses the same structure and functions for the processing of C before and after the land-use change and secondly, we need a model that required minimal inputs since there is a dearth of climate and/or weather data in most boreal location suitable for reservoir creation. FAQ-DNDC combines

mechanistic components of C processing and transport in water-saturated soils and the overlying water column to predict $CO_2$ emissions from reservoirs. FAQ-DNDC runs of a daily time-step and requires minimal inputs. After presenting the development of FAQ-DNDC we test its performance against observational data including a multi-year (2006–2012) eddy covariance (EC) record of $CO_2$ fluxes and other variables such as water column respiration and sedimentation rate for the newly created boreal reservoir in northern Quebec, the Eastmain-1 reservoir. We examined

the inter-annual and seasonal changes of $CO_2$ emissions in response to the environmental factors.

      The objectives of this paper were to (1) describe the scientific foundation, mathematical formulation, and major assumptions of the FAQ-DNDC reservoir model, (2) demonstrate and access seasonal and inter-annual variability in $CO_2$ emissions, and (3) evaluate how and by what mechanisms, flooding alters the C fluxes across air−water interface. Based on limited empirical data, we test the hypothesis that the boreal reservoir will be a net source of $CO_2$ to the

atmosphere. We further hypothesize that the exchanges will be the largest in the first one to two decades and will then show little secular change thereafter—i.e. year-to-year variability around a fairly constant mean and that environmental factors such as air temperature and wind speed can regulate reservoir C dynamics by affecting the physical and biogeochemical processes and their interactions.

## 2 Materials and methods

### 2.1 Model description

      The FAQ-DNDC model aims to minimize the climatic inputs and required parameters, while at the same time capture the well-understood key physical and biogeochemical processes such as water−sediment C exchange and air−





water gas exchange. It can be applied for any lakes and reservoirs at any geographical location with different soil and climate conditions, but is currently parameterized for northern boreal reservoirs. FAQ-DNDC has three major sub-models (Fig. 1): (1) a water column C processing sub-model that simulate organic and inorganic C dynamics in the water column overlying the sediment loosely based on Hanson et al. (2004); (2) a thermal and water stratification sub-model that calculates water and sediment temperatures, water vertical movement, and ice-cover duration (Snow, Ice, WAter, and Sediment: SIWAS, Wang et al., 2016); and (3) a sediment biogeochemical sub-model that is able to simulate anaerobic biogeochemical processes (Forest-DNDC, Li et al., 2005;Li et al., 2000). The key processes and the linkage among sub-models were described as follows.

### 2.1.1 C processing in the water column

We developed the water column C model based on Hanson et al. (2004). As in Hanson et al. (2004), POC includes both living POC ($POC_L$) and dead POC ($POC_D$) in the water column. It has been widely accepted that DOC in rivers and lakes spans a range from very labile to very refractory (Amon and Benner, 1996;Hanson et al., 2015;Søndergaard and Middelboe, 1995), therefore two kinetically distinct DOC pools: labile and refractory, are used to describe the fate of DOC in the water column. Dissolved inorganic C (DIC) is the sum of the chemical species including dissolved $CO_2$, bicarbonate ($HCO_3^-$), and carbonate ($CO_3^{2-}$).

If thermal stratification (i.e., epilimnion and hypolimnion) develops during the open water period, both epilimnion and hypolimnion have four organic C pools (i.e., $POC_L$, $POC_D$, $DOC_L$, and $DOC_R$) and one DIC pool each. Unlike Hanson et al. (2004), our model includes algorithms that simulate the vertical mixing of C during the ice free season by incorporating the SIWAS thermal and water stratification sub-model (see Section 2.1.2). The water column has a single layer—hypolimnion when the reservoir is ice-covered.

Inflow and outflow, inputs and exports DOC, DIC, and POC to and from the water column of a reservoir. The concentrations of the C species in the outflow are determined by their concentrations in the reservoir water column. DOC and DIC in precipitation are added to the water column, as atmospheric DOC and DIC deposition with precipitation could contribute up to 13% and 8% of total DOC and DIC loading to lakes, respectively (Dillon and Molot, 1997). This source of DOC is assumed to be refractory (Moore, 2003). There are no C species (e.g., DOC and DIC) existing in the solid (i.e., snow and ice) and gas (vapor) forms of water exchange in the model. Also we assume that at present there is no dry deposition of C although there could be short time inputs in the boreal region if there are wildfires in close proximity of a reservoir.

In our water column C sub-model, plankton gross primary production (*GPP*) and respiration (*PR*) (mg C m$^{-3}$ day$^{-1}$) are functions of available light, water temperature ($T_w$, °C), chlorophyll-*a* concentration (*Chla*, µg L$^{-1}$), DOC concentration ($DOC_{conc}$, mg C L$^{-1}$), and mixed depth ($z_{mix}$, m) (Pace and Prairie, 2007;Carignan et al., 2000):

$$GPP = 10^{0.80-0.67\log(z_{mix})+0.75\log(Chla)+1.33\log(T_w)-0.77\log(DOC_{conc})} \tag{1}$$

$$PR = 10^{0.67-0.94\log(z_{mix})+0.77\log(Chla)+1.28\log(T_w)-0.64\log(DOC_{conc})} \tag{2}$$





where $z_{mix} \leq z_{photic}$, $z_{photic}$ is the sunlight zone depth. *Chla* in photic zone is estimated by using a function of $POC_L$ concentration ($POC_{L,conc}$, mg C L$^{-1}$) and given by Desortová (1981):

$$chla = 1.58 + 4.97 POC_{L,conc} \qquad (3)$$

The exudation of plankton ($R_{EXU}$) from GPP to DOC is estimated as 10% of GPP. The ratio between labile and refractory components is assumed to be 7:3 (Hanson et al., 2004;Cole et al., 2002). The difference between the remaining GPP* (GPP–$R_{EXU}$) and PR goes to the growth of plankton biomass. $POC_L$ mortality is estimated as 0.03 day$^{-1}$ and 0.90 day$^{-1}$ for epilimnion and hypolimnion, respectively (Hanson et al., 2004).

POC sedimentation is dependent on the settling time ($t_{fall,h}$, day) and reservoir water residence time ($t_r$, day) (Walker, 2001). In this case, the sedimentation fraction ($f_{s,h}$) is determined as:

$$f_{s,h} = t_{fall,h} / t_r \qquad (4)$$

where $t_{fall, h}$ is determined by layer height (m) and sinking velocity ($W_h$, m s$^{-1}$). Sinking velocity is calculated using the Stokes equation that is restricted to use under non-turbulent conditions.

$$W_h = \frac{ESD^2 g \Delta\rho}{18u} \qquad (5)$$

where *ESD* (m) is equivalent spherical diameter and is set at 18 μm for POC (Ruiz et al., 2004); g (9.81, m s$^{-2}$) is the gravity acceleration; $\Delta\rho$ is the difference of density between POC and water (100 kg m$^{-3}$; Ruiz *et al.*, 2004); and *u* (kg m$^{-1}$ s$^{-1}$) is dynamic viscosity that is estimated by an exponential function of water temperature derived from Kalff (2002). Given the drag coefficient under turbulent conditions has limited effects on small particles (Kirillin et al., 2012), we estimate POC sinking velocity in epilimnion where turbulent conditions exist using Eq. (5).

First order kinetic function is used to characterize the decay rate of organic C, and the decay rates are temperature dependent and adjusted using a $Q_{10}$ function. The $CO_2$ production from organic C decomposition is added to the DIC pool. The fraction ($Fr_{co_2}$) of dissolved $CO_2$ in the DIC pool is calculated as a function of *pH*, following Kalff (2002):

$$Fr_{co_2} = \begin{cases} -0.087pH^2 + 0.71pH - 0.453 & 4 < pH < 6.5 \\ 0.5 & pH = 6.5 \\ 0.152pH^2 - 2.529pH + 10.512 & 6.5 < pH < 8.4 \end{cases} \qquad (6)$$

Gas exchange flux ($F_{CO_2}$, mmol m$^{-2}$day$^{-1}$) across air–water interface is estimated using Fick's law and is given as:

$$F_{CO_2} = \frac{k_{CO_2}(pCO_{2\,water} - pCO_{2\,air})}{K_H(T)} \qquad (7)$$

where $k_{CO_2}$ is $CO_2$ diffusion coefficient (m d$^{-1}$) and is less than $z_{mix}$ that is calculated in the thermal sub-model; $pCO_{2\,water}$ and $pCO_{2\,air}$ are the partial pressure $CO_2$ (μatm) in the surface water and the atmosphere, respectively; $K_H$ (*T*) is Henry's volatility constant ([m$^3$ atm] mol$^{-1}$) for $CO_2$ at a given temperature (*T*, K) according to Sander (2015):



$$K_H(T) = \frac{9.86 \times 10^{-6}}{H(T)} \tag{8}$$

$$H(T) = H^{\ominus} \times \exp(C(\frac{1}{T} - \frac{1}{T^{\ominus}})) \tag{9}$$

where $H(T)$ is Henry's solubility constant at a given T; $H^{\ominus}$ is the H at a standard temperature $T^{\ominus}$ (283.15 K); C is the temperature dependency of $H$. $k_{CO_2}$ is related to the piston velocity ($k_{600}$, m d$^{-1}$) and the Schmidt number of $CO_2$ ($Sc_{CO_2}$), and given by:

$$k_{CO_2} = k_{600}(600/Sc_{CO_2})^n \tag{10}$$

where the constant of 600 is the Schmidt number of $CO_2$ at 20 ℃ in freshwater; n is 0.50 and 0.33 in the low (≤3 m s$^{-1}$) and high (>3 m s$^{-1}$) wind conditions, respectively. $k_{600}$ is estimated as in Vachon and Prairie (2013):

$$k_{600} = 2.51 + 1.48\,U_{10} + 0.39 U_{10} \log_{10} LA \tag{11}$$

where $U_{10}$ is the wind speed (m s$^{-1}$) at 10-meter height; $LA$ is the lake/reservoir surface area (km$^2$). $Sc_{CO_2}$ is calculated using water surface temperature ($T_{ws}$, ℃) and given by (Wanninkhof, 1992):

$$Sc_{CO_2} = 1911.1 - 118.11 T_{ws} + 3.4527 T_{ws}^2 + 0.04132 T_{ws}^3 \tag{12}$$

We assume that $pCO_{2\,air}$ is 380 µatm; $pCO_{2\,water}$ is estimated as

$$pCO_{2\,water} = 83333.3 K_H(T_{epi}) DIC_{conc} Fr_{co_2} \tag{13}$$

where $DIC_{conc}$ is the DIC concentration (mg C L$^{-1}$) in epilimnion.

### 2.1.2 Thermal dynamic and water mixing

The newly developed one-dimension (1-D) lake thermal dynamic model, SIWAS (Wang et al., 2016), is incorporated into FAQ-DNDC for simulating the effects of ice-cover timing, temperature, and vertical mixing on the reservoir C dynamics. SIWAS is designed to bridge the gap between physical and biogeochemical processes in remote boreal regions with limited climate data. The sub-model requires daily climatic inputs (e.g., air temperature, wind speed, and relative humidity) that are commonly available, and are also used by other sub-models of FAQ-DNDC. If daily inflow and outflow data are available, the sub-model is able to simulate the fluctuations in the water surface elevation due to dam operations.

SIWAS includes six subroutines: i.e., surface energy balance, water mass balance, heat transmission and diffusion, snowpack, ice physics and dynamics, water vertical mixing, and sediment thermal dynamics. Solar radiation is estimated from the function in Forest-DNDC (Li et al., 1992) based on latitude, day of year, a constant solar radiation into atmosphere (1370W m$^{-2}$), and a constant cloud cover ($Cc$) of 0.47. Albedo changes with reservoir surface cover (i.e., snow, ice, and water), snow depth, and ice depth (Duguay et al., 2003). Incoming long-wave radiation is determined using a power function of the specific humidity (Rosa and Stanhill, 2014), while outgoing long-wave radiation from the reservoir is calculated using the Stefan-



Boltzmann law. Sensible and latent heat is estimated using a modified bulk aerodynamic method that dynamically computes heat transfer coefficient and surface roughness length (Verburg and Antenucci, 2010), and can account for snow/ice sublimation. The processes of snow melting, ice growth and decay are simulated based on residual energy (Wang et al., 2016).

Water mass balance is determined by the difference between precipitation and inflow, and the outputs of evaporation and outflow. Light transmission is quantified using the Beer-Lambert law. Heat diffusion is solved using finite differences. Turbulent kinetic energy (TKE) algorithm that accounts for kinetic energy of convective, stirring, and shearing is used to simulate surface mixing regimes. Soil thermal properties are calculated based on the fraction of different soil component (i.e., mineral, organic matter, and water) thermal properties.

### 2.1.3 Sediment C dynamics

To simulate the effects of flooding on soil biogeochemical processes, we modified the Forest-DNDC model that can simulate processes responsible for the production, consumption, and transport of greenhouse gases (e.g., $CO_2$ and methane [$CH_4$]) in forest and wetland soils. These processes are typically mediated by microbes and simulated according to laws in chemistry, biology, and kinetics in the model. The DNDC model is able to model different biogeochemical processes in soils where aerobic and anaerobic microsites exists simultaneously or alternately using the 'anaerobic balloon' concept (Li, 2000). This model is a well-studied model that has been broadly applied in agriculture, forests, and wetlands for estimating $CO_2$, $CH_4$, and nitrous oxide ($N_2O$) fluxes from soils during the last decades (see review by Gilhespy et al., 2014). The version of forests and wetlands, Forest-DNDC (Li et al., 2005;Li et al., 2000;Zhang et al., 2002), is used in this study. The Forest-DNDC model has been parameterized and tested for boreal and temperate forest and wetland ecosystems (e.g., Kim et al., 2014a;Webster et al., 2013). It thus is especially suitable for use under flooded conditions.

Similar to Forest-DNDC, soil organic matter is divided into four organic matter pools (Fig. 1): litter, microbial biomass, active humus, and passive humus in FAQ-DNDC. Each pool except the passive humus has labile and resistant components. The soil is vertically separated into organic and mineral layers with different soil characteristics (e.g., bulk density and clay content) according to litter and soil types. Terrestrial plants die on submergence. The C biomass including foliage, woods, and roots is then input to the litter pool once water depth is equal to mean water depth. After flooding, the input of organic C to the sediment is through POC sedimentation described in Section 2.1.1, and then the fresh organic C is allocated to the very liable, labile, and resistant litter pools based on the C: N ratio of POC.

Sediment organic C decomposition is simulated based on decay rates that are modified by temperature and redox potential (Zhang et al., 2002). Oxygen in the fully water-saturated sediment will be depleted, and hence the partition of decomposition production of DOC and $CO_2$ will be changed once flooding events occur. An empirical parameter of oxygen effect ($f_{O_2}$) is used for partitioning decomposition products of $CO_2$ and DOC for litter C pools. At higher $f_{O_2}$, higher percent of decomposition products is considered to be $CO_2$, whereas the remaining goes to the DOC pools. Previous incubation experiments showed that anaerobic conditions can increase DOC concentrations for flooded boreal forest organic soils and





peats (Kim et al., 2014b), but reduce $CO_2$ production due to the lack of oxygen (Kim et al., 2015). The impact of this assumption on the model is explored through sensitivity analysis of this parameter.

$CO_2$ produced in the terrestrial soil is directly released to the atmosphere in Forest-DNDC, while in FAQ-DNDC, $CO_2$ produced in the sediment is stored in sediment pore water with the form of DIC. DOC leaching in the terrestrial soil is
simulated in the Forest-DNDC model, while our model estimates infiltration or seepage at the sediment–water interface. Solute (i.e., DIC and DOC) diffusion within sediment is calculated using Fick's second law:

$$\frac{\partial C}{\partial t} = D_{eff,z} \frac{\partial C^2}{\partial z^2} \tag{14}$$

where $C$ is the solute concentration (g C m$^{-3}$) and $D_{eff, z}$ is the effective diffusion coefficient (m$^2$ s$^{-1}$) at depth $z$ (m). The effective diffusion coefficient is the sum of diffusion processes involved in the vertical transport of solutes in sediments. It can
be estimated by (Portielje and Lijklema, 1999):

$$D_{eff,z} = D_m^* + D_{tur}e^{-kz} \tag{15}$$

where $D_m^*$ is the effective molecular diffusion coefficient ($10^{-9}$ m$^2$ s$^{-1}$), $D_{tur}$ is the turbulent diffusion coefficient ($10^{-9}$ m$^2$ s$^{-1}$) decaying exponentially with sediment depth, and $k$ is a the attenuation coefficient. The effective molecular diffusion coefficient of DIC ($D_{m,DIC}^*$, $10^{-9}$ m$^2$ s$^{-1}$) is calculated using a function of temperature derived from Zeebe (2011) and given as:

$$D_{m,DIC_i}^* = 0.06T_{sed_i} - 17.0 \tag{16}$$

where $T_{sed}$ is the layer temperature (K), $i$ is the $i$th sediment layer. Given the weakly bound organic C between the sediment and the interstitial water, the effective molecular diffusivity ($D_{m,DOC}^*$, $10^{-9}$ m$^2$ s$^{-1}$) of DOC in the sediment is amended based on the molecular diffusion coefficient in the pore water ($D_{m,DOC}$, $10^{-9}$ m$^2$ s$^{-1}$), a partition coefficient for sorption and desorption ($K_r$, L water kg$^{-1}$ sediment), and sediment physical characteristics such as bulk density ($\rho_b$, g cm$^{-3}$) and porosity ($\varepsilon$, cm$^3$ water
cm$^{-3}$) as follows (Thoma et al., 1991):

$$D_{m,DOC}^* = \frac{D_{m,DOC}\epsilon^{4/3}}{\epsilon + \rho_b K_r} \tag{17}$$

The $\rho_b$, $\varepsilon$, and $K_r$ are provided in each time-step by the thermal and water mixing sub-model described in Section 2.1.2. The diffusion within the sediment is solved using finite differences. Sediment-water exchange flux of solute ($F$, g C m$^{-2}$ day$^{-1}$) is calculated using Fick's first law of diffusion:

$$F = D_{eff,0}\left(C_{pw} - C_{wb}\right)/d \tag{18}$$

where $C_{pw}$ and $C_{wb}$ are the concentration of solute (g C m$^{-3}$) in the pore water of sediment surface layer and the overlying water layer, respectively; $d$ is the distance (m) between midpoints of bottom water layer and top sediment layer.



## 2.2 Study site and data collection

The Eastmain-1 reservoir (EM-1, 51 to 52 °N and 72 to 76 °W) in northern Quebec, Canada was constructed at the end of 2005 resulting from the damming of the Eastmain River. The full reservoir has a surface area of 623 km$^2$ and a total storage capacity of 6.94 km$^3$. The surface elevation varies ~9 m over the reservoir operations. The mean depth of the reservoir is 11 m. The EM-1 power complex can generates 1248 megawatt hours of electricity. The Eastmain River has an average discharge of 635 m$^3$ s$^{-1}$. The EM-1 reservoir area has a continental climate with mean annual temperature of −1.5˚C (daily maximum and minimum temperature of 20.4 and –27˚C) and mean annual precipitation of 969 mm, with 32% falling as snow (measured for 15 years between 1981 and 2010 at Bonnard weather station, 50.73°N 71.05°W, http://climate.weather.gc.ca).

Pre-flooded landscapes were composed of forests, wetlands, lakes, and rivers. Black spruce (*Picea mariana* Mill. BSP) forests covered an area of 296 km$^2$, or ~50% of the pre-flooded landscape (Teodoru et al., 2012). The groundcover was dominated by bryophytes and lichens (Paré et al., 2011, Ullah et al., 2009). Soil texture was sandy loam and organic soil layers were typically 15-40 cm thick (Bergeron et al., 2007, Ullah et al., 2009). The site characteristics of pre- and post-flooded landscapes are summarized in Table 1.

Air temperature, relative humidity, and wind speed data over a 5-yr period between March, 2007 and October 2012 were obtained from an EC tower (52.12°N, 75.93°W) established on an island in the middle of the reservoir (Strachan et al., 2016). We generated daily means from the original half-hour measurements. The daily precipitation data were collected at one of the closest weather stations (Chibougamau Chapais A, 49.77°N, 74.53°W, http://climate.weather.gc.ca) where daily precipitation data are available. The reservoir's inflow and outflow data from 2007 to 2012 were obtained from Hydro-Quebec.

To test our model, we collected $CO_2$ flux data measured by EC from July 2006 to October 2012. Details on EC measurements, data processing, and quality control were described elsewhere (Strachan et al., 2016;Lemieux, 2011). The average data coverage during the open water period was only 25%: the largest amount of missing data is for spring and daily in the morning. We also had access to mean $CO_2$ fluxes from floating chambers, water column respiration, POC sinking rate, and the mean concentrations of DOC, DIC, and chlorophyll-*a* for flooded forest sites during the open water periods from 2006 to 2008 (Teodoru et al., 2011;Teodoru et al., 2013). Although we do not have access to the direct and continuous measurement on surface water *pCO₂* nearby the tower location, we compared our modeled mean *pCO₂* with outflow *pCO₂* measured in the EM-1 generation station.

## 2.3. Parameterization, calibration, testing, and evaluation

The DIC concentrations from the Eastmain River vary from 0.33 to 0.42 mg C L$^{-1}$ and are relatively constant during the open water period (Teodoru et al., 2009). A mean value of 0.37 mg C L$^{-1}$ for DIC concentration was used in our simulations (Table 2). DOC concentration from the river was set to 7.99 mg C L$^{-1}$ which is the mean value of the observed range of the region between 7.51 and 8.38 mg C L$^{-1}$ (Teodoru et al., 2009). We estimated total POC concentration to be 0.7 mg C L$^{-1}$ using the ratio (4.7–14.6) of DOC:POC export for boreal rivers (Hope et al., 1994), yielding an estimate between 0.5 and 1.8 mg C





L$^{-1}$. POC$_L$ concentration of 0.1 mg C L$^{-1}$ in inflow was estimated using observed chlorophyll-*a* concentration (1.58–2.78 µg L$^{-1}$) from natural lakes in Northern Quebec (Teodoru et al., 2013) and the empirical relationship between POC$_L$ and chlorophyll-*a* concentrations following Eq. (3). DIC concentration in precipitation of 0.6 mg C L$^{-1}$ appears to be reasonable in the boreal region. A value of 2 mg C L$^{-1}$ for DOC concentration in precipitation was used in this study (Moore, 2003).

Terrestrially-derived DOC$_L$ in inflow is estimated to be 19% of total DOC (Søndergaard and Middelboe, 1995). A higher percentage (60%) of total DOC from sediment export to the water column is assumed to be the labile component, as autochthonous DOC has been thought of more bioavailable than DOC receiving from surrounding catchments (Koehler et al., 2012); the remaining DOC is assumed to be refractory. The decay rate of DOC$_L$ and DOC$_R$ at 20 °C is set at 0.1 and 0.005, respectively (Søndergaard and Middelboe, 1995). POC$_D$ has a relatively high decay rate of 0.05 in our simulations (Hanson et

al., 2004). The C:N ratio in POC$_D$ is estimated to be 16.0 within the range (10.7-25.5) reported for boreal lakes and reservoirs (Teodoru et al., 2013), resulting in very liable litter input through POC sedimentation. The decomposition rates of the sediment C pools at 20°C were parameterized for the forest soils in two boreal black spruce forests in Northern Quebec (Kim et al., 2014a) (Table 2).

    The model was run with a daily time-step and a vertical resolution of 0.02, 0.01, 0.1, and 0.05 m for snow, ice, water,

and sediment, respectively. We filled the gaps of the input data by using a 4-yr (2008–2011) average daily time series. The inundation was assumed to occur on November 1, 2015. The initial woody vegetation C biomass in foliage, woods, and roots and soil organic C in the organic and mineral layers were obtained from literature (Table 1), for this model development and evaluation. In an operational mode the initial C stores would be simulated with a pre-flood version of Forest-DNDC (Kim et al. 2014a). FAQ-DNDC then created a water column overlying the forest soil by allowing an inflow and 100 m$^3$ s$^{-1}$ of outflow,

until the water depth reached the mean depth. Once the reservoir was full, we assumed the vegetation C biomass was added to the litter C pools. Most tree stems were removed through reservoir surface elevation change due to dam operation in winter. Here we estimated 70% of wood C removal ($R_w$) in the winter of 2005 for the test run. The impact of this assumption on reservoir CO$_2$ emissions was explored using sensitivity analysis by varying the fraction of stems removal.

    We calibrated the model using lake indices: concentrations of DIC and DOC, measured during the ice-free period of

the year 2006, 2007, and 2008. We then tested the model simulation results against CO$_2$ fluxes measured from the EC tower. We expected that simulated CO$_2$ fluxes, water column respirations, and POC sedimentation to fall within measured range. We evaluated the performance of FAQ-DNDC in two phases: warming and cooling, given that ice-off and -on dates greatly affect CO$_2$ evasion timing, and thereby influence the overall model performance.

    Several approaches were used to quantitatively evaluate the model performance. We computed root mean square error

(RMSE), comprising to components: systematic RMSE$_S$ and unsystematic or random RMSE$_U$ error, the refined Willmott index (Willmott et al., 2012), and Pearson correlation coefficients. The refined Willmott index of agreement is an index of model efficiency and varies between −1 and 1: a value closer to 1 indicates the better model performance. For the Pearson's correlation coefficient, a value of 1 indicates strong correlation between observations and model output, while values near 0 indicate weaker and often insignificant correlations between observations and model output.



### 2.4. Sensitivity analysis and simulation experiments

We performed sensitivity analysis by assessing the magnitude of change in $CO_2$ evasions in response to changes in the magnitude of $R_w$, $f_{O_2}$, and two climate forcings (i.e., air temperature and wind speed). We changed $R_w$ (0.4 and 0.8, compared to the 0.6 used in the model test) and $f_{O_2}$ (0.4 and 0.8, compared to 0.6) to explore the model uncertainty due to flooding events. We increased and decreased the air temperature by 2 °C and wind speed by 20% compared to that in the baseline simulation to examine the model sensitivity while keeping the remaining parameters and inputs at their original values. The sensitivity of temperature was due to two interactions: physical attributes such as ice duration, and biogeochemical attributes such as in the rate of decomposition. For wind speeds, we wanted to examine whether the $CO_2$ emissions are ultimately controlled by $CO_2$ production in the reservoir or the environmental factor (i.e., wind speeds) that control the gas exchange coefficient across air–water interface (see Eqs. [7–13]). To assess the effects of these changes on $CO_2$ emissions over the engineering lifetime of a reservoir, we ran FAQ-DNDC repeating the mean climate and input parameters for 100 years. The model was considered to be sensitive to the parameters or climate input if the mean change exceeded 10% of the base run.

### 3 Results

### 3.1 Model performance

Overall, simulated $CO_2$ fluxes generally follow the pattern of, and are consistent with, the EC $CO_2$ flux observations (Fig. 2). The simulated $CO_2$ fluxes ranged from a low of 0.6 to a high of 9.2 g C m$^{-2}$ d$^{-1}$ while the RMSE, RMSE$_S$, and RMSE$_U$ of the simulated and observed daily $CO_2$ flux were in the range of 0.6–1.3, 0.2–1.1, and 0.4–1.3 g C m$^{-2}$ d$^{-1}$ for the open water period, respectively. RMSE$_S$ was larger than RMSE$_U$ in the year of 2006, 2008, and 2009. The $d_r$ was in the range of 0.32–0.56 across the simulation period with the exception of 2012 ($d_r = 0.20$), and the Pearson's correlation coefficients were 0.32–0.55 with the exception of 2006 ($r = -0.09$) and 2010 ($r = 0.08$). Overall, simulated mean $CO_2$ fluxes were similar to EC observations over the period of 2006–2012, but both EC measured and modelled $CO_2$ fluxes were much smaller than chamber-based measurements (Fig. 3).

FAQ-DNDC reasonably predicted POC sedimentation rate and concentrations of DOC, DIC, chlorophyll-$a$, but tended to underestimate water column respirations (Fig. 4). Specifically, the model accurately predicted POC sinking rate for the open water period of 2008. However, simulations underestimated water column respiration by 22–31%, for the first three years (2006–2008). The model predicted relatively stable chlorophyll-$a$ concentrations compared to the observations. The modeled mean $pCO_2$ generally followed the observed annual and seasonal change in outflow $pCO_2$ (Fig. S1).

### 3.2 Annual $CO_2$ emissions with reservoir age

Both measured and modeled annual daily mean $CO_2$ fluxes over the observation period (2006–2012) showed a decreasing trend with the years (2006–2008) after flooding (Fig. 3), but the model predicted relatively high emission rate for



2011, and 2012, respectively. Scenario simulations showed the steep decline in annual $CO_2$ emissions occurred in the first three years (~205 to ~120 g C $m^{-2}$ $yr^{-1}$), while during the rest of reservoir lifetime modelled fluxes declined slowly to ~114 g C $m^{-2}$ $yr^{-1}$ (Fig. 5). The modeled decline was associated with the decreasing benthic fluxes (dissolved $CO_2$) that initially contributed as high as 37% of the annual $CO_2$ emissions, and declined gradually thereafter as low as ~30% in the old reservoir.

## 3.2 Seasonal exchange of $CO_2$ emissions

The modeled peak $CO_2$ emissions (3.2–9.2 g C $m^{-2}$ $day^{-1}$) from the EM-1 reservoir typically occurred in the first week after ice cover break up (DOY from 120 to 150 over 2006–2012, Fig. 2). Daily $CO_2$ emissions after break up generally decreased, but remained above zero during the summer. With water mixing in late summer (i.e., the period of cooling, DOY 210–240), modeled daily $CO_2$ emission rates reached a second peak (1.8–3.2 g C $m^{-2}$ $day^{-1}$) but much smaller compared to the spring peak. Then daily $CO_2$ fluxes decreased and became ~0.7 g C $m^{-2}$ $day^{-1}$ during the rest of open water period. In the ice-cover period, there was no gas exchange across air–water interface.

## 3.4 Sensitivity analysis

The modeled annual $CO_2$ emissions and benthic dissolved $CO_2$ fluxes showed great sensitivity to $Rw$, $f_{O_2}$ and air temperature, but not to wind speed. (Fig. 5). Decreasing the $Rw$ to 0.4 from the 0.6 used in the base simulation led to an increase of the annual $CO_2$ emissions by up to 10% in the first 3 years, and then the increase subsequently declined to ~2%. The opposite patterns occurred when $Rw$ was increased to 0.8. The simulated response to the change in $f_{O_2}$ was linear. By increasing $f_{O_2}$ to 0.8, the dissolved $CO_2$ fluxes (74% of DIC fluxes) across sediment–water interface increased by 21±4% over the simulation period, and the annual $CO_2$ emissions increased by 6±2%. The warmer climate scenario (+2 °C) caused an increase in dissolved $CO_2$ fluxes across sediment–water interface by approximately 50% in the early stage of the reservoir, and then the increase rate generally declined to 2% over 100 years. The $CO_2$ emission initially increased up to 43%, and then the increase rate declined to 12% with reservoir aging. Contrastingly, the cooler climate scenario (–2 °C) resulted in a decline for both C fluxes across sediment–water and air–water interfaces. The decreasing rates for benthic fluxes and $CO_2$ effluxes declined to 6% and 9% from 20% and 16%, respectively. Generally, changes to the wind speed influenced the C fluxes across the sediment–water and air–water interface, but not significantly. Both increasing and decreasing wind speeds enhanced annual $CO_2$ emissions only by 1 and 1% over 100 years, respectively. Benthic dissolved $CO_2$ fluxes increased by 1.5 % for the higher wind speed scenario and 0.1% for the lower wind speed scenario.

Besides the magnitudes, $Rw$ and $f_{O_2}$ had limited effects on the seasonal change in $CO_2$ emissions, while air temperatures and wind speeds greatly influenced the seasonal pattern (Fig. 6). Increasing air temperature by 2 °C made an early and grater spring emission peak (6.0 *vs.* 5.3 g C $m^{-2}$ $day^{-1}$) and a longer emission period (i.e., the open water length, 200 *vs.* 182 days) over the period of 2006 to 2012, while a lower spring emission peak (4.4 g C $m^{-2}$ $day^{-1}$) and shorter emission period (168 days) occurred under the cooler climate scenario. Higher wind speeds led to a higher spring emission peak (6.0 g





C m$^{-2}$ day$^{-1}$) but slightly shortened emission period by 1 day; lower wind speeds caused a longer emission period (187 days), but reduced the mean daily fluxes (0.84 *vs.* 0.88 g C m$^{-2}$ day$^{-1}$). Both environmental variables had limited effects on the magnitude of CO$_2$ emissions during autumn and summer.

## 4 Discussion

### 4.1 Effects of flooding on C dynamics

According to our simulations the EM-1 reservoir is a net CO$_2$ source to the atmosphere over its lifetime (Fig. 5), which supports our first hypothesis. Our finding agrees with previous observational studies that examined GHG fluxes in boreal reservoirs (Teodoru et al., 2012). Besides terrestrially-derived DOC from surrounding catchment, mineralization of flooded terrestrial organic matter is an important regulator of C processing in young reservoirs (Venkiteswaran et al., 2013;Brothers et al., 2012b). Similar with power or exponential declines in annual CO$_2$ emissions reported in the global syntheses (St. Louis et al., 2000;Barros et al., 2011), the modeled change supports our second hypothesis that the gas exchange is initially high, gradually decline, and then become relatively flat (but still decreasing) with reservoir age. Our finding is also largely consistent with the empirical study reporting a first-order exponential decay trend using data from the EM-1 reservoir and other older boreal reservoirs (Teodoru et al., 2012). However, the empirical model estimated longer decline period (12 to 15 years) than our simulated (three years) using the process-based model. A previous modelling study without the consideration of the water column C processing and transport reported that the fast decline occurred for the first four decades after flooding at the boreal forest site (Kim et al., 2016). Our modeled annual change is attributed to not only the benthic processes such as relatively low decomposition rates regulated by temperatures and redox potentials under anaerobic conditions, but also the water column processes.

Our simulations also show that sediment organic C keeps loosing over the simulation period (i.e., 100 years) (Fig. 7). Here, the organic C burial efficiency was defined as the ratio between the sum of DOC and DIC fluxes and the POC sinking rate. Unlike the forest ecosystems, both physical (e.g., sedimentation and diffusion) and biogeochemical (e.g., decomposition) processes in the water column and sediments co-determine reservoir C dynamics. Large amounts of terrestrial organic matter in reservoir sediments provides a continuous source of CO$_2$, while the overlying water column, like blanket, slows down CO$_2$ escaping from the sediment to the atmosphere. We did not incorporate methane production in our estimates yet, if methane production is included, the reservoir would probably need more time than reported here.

Flooded terrestrial organic C, ranging from 10.6 to 12.9 kg C m$^{-2}$ in our sensitivity analysis, positively affect CO$_2$ emissions from the reservoir (Fig. 5 a). This agrees with previous observation studies of reported that the spatial heterogeneity of surface CO$_2$ fluxes from the EM-1 reservoir is related to the former landscape types with different quantity of organic C flooded (Brothers et al., 2012a;Teodoru et al., 2011). However, Venkiteswaran et al. (2013) reported that the amount of flooded organic C (ranging from 3.1 to 4.6 kg C m$^{-2}$) had limited effects on GHG fluxes in





the first five years after flooding from their experimental reservoirs on the Canadian Shield. The different quantities of terrestrial organic C flooded between two studies might produce the discrepancy. We also argue that $CO_2$ emissions across air–water interface are not only controlled by $CO_2$ production that is related to organic C content in the flooded soils, but also physical transport processes such as air–water gas exchange (Eqs. [7–13]) and sediment–water mass

transport (Eqs. [15–18]).

The partitioning parameter $f_{O_2}$ for decomposition products: $CO_2$ and DOC, significantly alters benthic C fluxes from the sediment, but has weak effects on air–water gas exchange (Fig. 5b). Our results are largely consistent with laboratory incubation studies that have shown a negative influence of oxygen availability (aerobic vs. anaerobic conditions) on $CO_2$ production (Kim et al., 2015;Moore and Dalva, 2001) and a positive effect on DOC concentration (Kim et al., 2014b). Many

studies have also found that soil inundation may constrain aerobic respiration (e.g., Lewis et al., 2014;Sanchez-Andres et al., 2010), and thereby relatively increase DOC concentration (Moore et al., 2003). However, incubation studies have also observed that flooded soils may produce more $CO_2$ under non-flooded conditions than under flooded conditions, if there was no control on oxygen levels (e.g., Oelbermann and Schiff, 2010, 2008). Flooding makes more water to circulate through the substrate, not directly leading to aerobic and anaerobic conditions. For the relatively weaker effects of $f_{O_2}$ on $CO_2$ emissions

than on benthic fluxes, we lack the benthic flux measurement data to test our model. A possible explanation is that benthic fluxes occurs during the whole year, whereas the $CO_2$ emissions occurs only during the open water period. An alternative explanation may be indirect physical effects such as the gas transfer function and water vertical movements.

## 4.2 Environmental controls on reservoir $CO_2$ emissions

Environmental factors especially temperature greatly influence the reservoir C dynamics (Fig. 5c, d). Equation (1, 2,

7–10) and temperature sensitive parameters listed in Table 2 revealed that biogeochemical and physical processes (e.g., decomposition and air–water gas transfer) regulate the C processing and transport in the reservoir system. Warmer climate can shorten ice cover length and increase sediment and water temperatures (Wang et al., 2016;Yao et al., 2014), which probably results in a higher mineralization rate (Gudasz et al., 2010) and a longer emission period (Wang et al., 2016). The gas transfer rate is higher at higher wind speeds and higher temperatures (Eqs. [7–13]), as shown in other studies (Jonsson et al.,

2008;Vachon and Prairie, 2013). Hence, warmer climate and higher wind speeds would enhance annual $CO_2$ evasions from the reservoir surface, if there is enough dissolved $CO_2$ available in the surface water. However, wind speeds have limited effects on water and sediment temperatures (Wang et al., 2016), which causes no significant change in the supply of dissolved $CO_2$ to the water column. Moreover, lower wind speeds may prolong the open water period by influencing surface energy balance (Wang et al., 2016), which causes high uncertainties in the effects of wind speed on annual $CO_2$ emissions.

As expected, the seasonal change of $CO_2$ emissions is influenced by thermal dynamics (e.g., ice cover duration and water vertical movement) that response to environment factors such as air temperatures and wind speeds (Figs. 2 and 6). The observed and modeled peaks of $CO_2$ emission often occurred after ice-off dates in late spring and at autumn turnover that is due to cooling of reservoir in the late summer. This pattern is largely consistent with previous EC observations showing similar





seasonal dynamics in boreal lakes during the open water period, although the modeled and observed flux magnitudes are greater from young boreal reservoir than from boreal lakes (Huotari et al., 2011;Mammarella et al., 2015;Jonsson et al., 2008). In the simulations the spring $CO_2$ flux reflects the release of the accumulated dissolved $CO_2$ under ice-cover over the winter, which comes from the mineralization of organic matter in the water column and benthic respiration. Once ice-cover breaks,

DIC concentration declines very fast due to the high $pCO_2$ gradient between surface water and the atmosphere. Thus, the largest discrepancies between simulated and observed $CO_2$ flux occurred in spring and they are due the lack of accuracy in estimating actual timing of break up (Wang et al., 2016). The dissolved $CO_2$ accumulates in the deeper layers due to water stratification in summer. Autumn turnover mixes deeper $CO_2$-enriched water with surface water, resulting in the second high emissions. This mechanism represented in the model agrees with speculated explanation for high fluxes that occurred during

periods of convective mixing due to nighttime cooling of surface water (Eugster et al., 2003).

Low summer $CO_2$ emissions in our simulations are directly attributed to relatively low concentration of dissolved $CO_2$ after spring emissions (Figs. 2 and 6), and high gross primary production (uptake of $CO_2$) during summer (Eqs. [1–3]). The observed fluxes are also lower in the summer although we do not have the temporal resolution in the observations of epilimnetic DIC concentrations to confirm the simulated cause. Our finding is consistent with a study that found $CO_2$ fluxes

were negatively correlated to chlorophyll-$a$ concentrations on a monthly basis in a temperate lake (Shao et al., 2015). The influence of plankton on the summer $CO_2$ emission rate can also be explained by that higher emission rate occurred during night compared to day (e.g., Podgrajsek et al., 2015;Shao et al., 2015;Eugster et al., 2003), and productive lakes had lower $CO_2$ fluxes than unproductive lakes (e.g., Mammarella et al., 2015;Huotari et al., 2011). On the other hand, the dissolved $CO_2$ in the water surface layer can be filled up with the deepening of epilimnion layer and mineralization of organic C in the water

column to keep dissolved $CO_2$ concentrations in a relatively high level compared to that in the atmosphere. Hence, the water surface still acts as a weak C source during summer. However, Lake Erie acted as a small C sink during the summer, although acted as an annual source (Shao et al., 2015). Despite the different climate between study sites, flooded terrestrial organic matter may continuously provide the water column $CO_2$ in the boreal reservoir, resulting in the difference between artificial and natural ecosystems in summer $CO_2$ fluxes.

**4.3 Uncertainty and future work**

Here we present the first modeling study on reporting the $CO_2$ emissions from a boreal hydroelectric reservoir using a mechanistic model that provides an understanding of the effects of reservoir creation on seasonal and inter-annual variability of C dynamics. We also acknowledge that our model does not yet take into account several factors that are known to influence C processing in the sediment and the water column. Firstly, we have largely simplified the

decomposition process omitting the effects of dissolved oxygen. Our results could be biased by the partitioning parameter ($f_{O_2}$) on the decomposition production: $CO_2$ and DOC, as dissolved oxygen availability not only changes with water depth and ice cover dynamics but also affects microbial utilization of DOC to $CO_2$. Secondly, although the



model is able to produce methane from sediments, we are still developing the methane oxidation process by accounting for the oxygen cycle. Finally, we ignore aspects of carbonate equilibria that affects water $pH$ and $pCO_2$. We are currently refining FAQ-DNDC to account for these limitations of the current version, focusing on the oxygen cycle, methane processes, and aquatic chemistry. The net change in C exchange due to the creation of a northern reservoir can be

quantified by running FAQ-DNDC for pre- and post-flood landscapes. Furthermore, we need to better quantify lake/reservoir C budget using a process-based model, enhancing our understanding and prediction under projected climate change (Hanson et al., 2015).

## 5 Conclusions

In summary, FAQ-DNDC, a parsimonious data requirement model suitable for regions where there is a dearth

of possible input data, predicts $CO_2$ emissions from a newly created boreal hydroelectric reservoir reasonably well. In this study, we demonstrated the seasonal and inter-annual variability of reservoir surface $CO_2$ emissions and examined their underlying physical and biogeochemical mechanisms, which are consistent with observations and speculated mechanisms in empirical studies. The thermal dynamics to some extent control the seasonality of $CO_2$ fluxes across air–water interface. The amount of flooded terrestrial organic C positively influence the $CO_2$ emissions, because

flooded terrestrial organic C will slowly release to the atmosphere over several decades to centuries after flooding. The boreal reservoirs may act as a net C source over their lifetime, and the C emissions would quickly decline in the first three years after flooding and then decrease slowly for the reaming of reservoir lifetime. Our model provides a useful tool to investigate the effects of reservoir creation on C dynamics and would help hydro-power industry to evaluate its greenhouse gas contributions.

**Acknowledgements**

This study was supported by a Natural Sciences and Engineering Research Council of Canada Collaborative Research Grant (award no. CRDPJ 401579-10) and funds from Hydro-Québec to IS and NTR. We thank Tim Moore and Andrew Pinsonneault for helpful discussions, and Laura Lyon for her comments on the manuscript.



**Figure captions:**

Figure 1: The model structure of FAQ-DNDC. Rectangles indicate major state variables (e.g., pools); solid arrows represent matter or heat flows; black dashed arrows indicate effects; colored arrows indicate the linkage among the three sub-models (Modified from Li et al., 2000;Hanson et al., 2004). T represents temperature in the thermal dynamic sub-model; Epi and Hyp represent, respectively, epilimnion (upper water layer) and hypolimnion (lower water layer), where pools of dissolved organic carbon (DOC), dissolved inorganic carbon (DIC), living particular organic carbon (POCL), and dead POC (POCD) exist in the water column.

Figure 2: Modelled (lines) and measured (symbols) daily $CO_2$ fluxes at the Eastmain-1 reservoir from 2006–2012. Open red circles represent observations less than 24 half-hourly; solid black circles represent observations more than 24. Model performance was evaluated using root mean square error (RMSE) and its components (systematic RMSEs and unsystematic RMSEu), refined Winmott index ($d_r$), and Pearson correlation coefficients ($r$); n is the number of days when there are available measurements from EC tower.

Figure 3: Comparison of annual mean daily $CO_2$ emissions between modeled and measured using floating chamber (2006–2009) and eddy covariance (2006–2012) methods, respectively. Error bars represent standard deviations.

Figure 4: Comparison between measured and modeled concentrations of (a) dissolved inorganic C (DIC), (b) dissolved organic C (DOC), (c) chlorophyll-$a$, and (d) water column respiration (WCR), and (e) particular organic C (POC) sinking rate ($F_{POC}$) during the open water period of 2006–2008. Error bars represent standard deviations.

Figure 5: Sensitivity of annual $CO_2$ emissions and benthic respiration (dissolved $CO_2$) to changes in (a) aboveground C removal fraction ($R_w$) and (b) oxygen effects ($f_{O_2}$) under concurrent climate and hydro-thermal regime, and to changes in (c) air temperature ($T$) and (d) wind speed ($u$). The curves were smoothed using a moving average filter with a span of 5.

Figure 6: Sensitivity of seasonal changes in $CO_2$ emissions to changes in (a) air temperature ($T$) and (b) wind speed ($u$). For the sake of celerity, we chose the year of 2010. The curves were smoothed using a moving average filter with a span of 5.

Figure 7: Simulated sediment carbon burial efficiency (= [$F_{DOC}$ + $F_{DIC}$] / $F_{POC}$) under concurrent climate and hydro-thermal regime. $F_{DOC}$ and $F_{DIC}$ represent fluxes of dissolved organic (DOC) and inorganic (DIC) carbon, respectively, from sediments to the overlying water column if they are positive. Positive $F_{POC}$ represents the sedimentation rate of particular organic carbon (POC) from the water column to the sediment.





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



Table 1. Site characteristics before and after the impoundment.

| Description | Value | Source |
|---|---|---|
| ***Pre-flood*** | | |
| Latitude | 52.17 | Strachan et al. (2016) |
| Vegetation type | Mature black spruce | Strachan et al. (2016) |
| Type of forest floor | Mor | Kim et al. (2014a) |
| Type of mineral soil | Sandy loam | Kim et al. (2014a) |
| Thickness of organic layer (cm) | 20 | Bergeron et al. (2007) |
| Thickness of mineral soil (cm) | 80 | Assumption |
| Above-ground living biomass (kg C m$^{-2}$) | 4.5[a] | Bergeron et al. (2007) |
| Below-ground living biomass (kg C m$^{-2}$) | 1.6[a] | Bergeron et al. (2007) |
| pH in forest floor | 4.3 | Kim et al. (2014a) |
| pH in mineral soil | 5.4 | Kim et al. (2014a) |
| Soil organic C in organic soil (kg C m$^{-2}$) | 6.9 (3.2) | Paré et al. (2011) |
| Soil organic C in mineral soil (kg C m$^{-2}$) | 2.1 (0.6) | Paré et al. (2011) |
| ***Post-flood*** | | |
| pH in the water column | 6.0 (5.8–6.2) | Vachon and Prairie (2013) |
| Surface area (km$^2$) | 623 | Strachan et al. (2016) |
| Maximum depth of the reservoir (m) | 12 | Assumption |
| Height of wind speed measurement to the maximum water surface (m) | 15 | Strachan et al. (2016) |
| DOC concentration (mg L$^{-1}$) | 6.5 (6.3–6.7)[b] | Teodoru et al. (2011) |
| DIC concentration (mg L$^{-1}$) | 1.1 (1.0–1.3)[b] | Teodoru et al. (2011) |
| Chlorophyll *a* concentration (µg L$^{-1}$) | 2.9 (1.6–4.2)[b] | Teodoru et al. (2011) |
| Total phosphorus (µg L$^{-1}$) | 18.1 (15.2–21.8)[b] | Teodoru et al. (2011) |
| Photic zone (m) | 4.1 (3.8–4.3)[b] | Teodoru et al. (2011) |

[a] Values from a similar region of boreal forest in norther Quebec.

[b] Measurements of flooded mature forest ecosystems during the first three years (2006–2008) after flooding.



Table 2. Parameter description and values (range or mean±standard deviation) used in the model.

| Parameter | Description | Value | Source |
|---|---|---|---|
| *Land use change* | | | |
| $Mtree$ | Tree mortality | 1.0 | Assumption |
| $Rw$ | Aboveground C removal fraction | 0.6 | Calibrated |
| $f_{o_2}$ | $O_2$ effects on sediment decomposition production to $CO_2$ | 0.6 | Assumption |
| *Water column* | | | |
| $H^{\ominus}$ | Henry's solubility constant for $CO_2$ at standard temperature ($T^{\ominus}$ =298.15 K) ($10^{-4}$ mol [$m^3$ Pa]) | 3.4 (3.1–4.5) | Sander (2015) |
| $C$ | Temperature dependency of Henry's solubility constant for $CO_2$ (K) | 2400 (2200–2900) | Sander (2015) |
| $C_{doc}\_p$ | DOC concentration in precipitation (mg $L^{-1}$) | 2.0 (1–8) | Moore (2003) |
| $C_{dic}\_p$ | DIC concentration in precipitation (mg $L^{-1}$) | 0.6 (0.6–5.5) | Górka et al. (2011) |
| $C_{doc}\_in$ | DOC concentration in inflow (mg $L^{-1}$) | 8.0 (7.5-8.4) | Teodoru et al. (2009) |
| $C_{dic}\_in$ | DIC concentration in inflow (mg $L^{-1}$) | 0.37 (0.33-0.42) | Teodoru et al. (2009) |
| $fl\_in$ | Fraction of inflow DOC to liable DOC | 0.15 (0.19±0.16) | Søndergaard and Middelboe (1995) |
| $C_{pocl}\_in$ | Living POC concentration in inflow (mg $L^{-1}$) | 0.1 (0–0.24)[a] | Teodoru et al. (2013) |
| $C_{pocd}\_in$ | Dead POC concentration in inflow (mg $L^{-1}$) | 1.0 (0.4–1.8)[b] | Hope et al. (1994) |
| $fr$ | Exclude ratio of GPP to resistant DOC | 0.03[c] | Cole et al. (2002) |
| $fl$ | Exclude ratio of GPP to labile DOC | 0.07[c] | Cole et al. (2002) |
| $ed$ | Algae mortality in epilimnion | 0.03 | Connolly and Coffin (1995) |
| $hd$ | Algae mortality in hypolimnion ($day^{-1}$) | 0.9 | Hanson et al. (2004) |
| $kpoc$ | Decomposition rate of dead POC at 20 °C ($day^{-1}$) | 0.05 | Connolly and Coffin (1995) |
| $kdocr$ | Decomposition rate of resistant DOC at 20 °C ($day^{-1}$) | 0.0055 (0.0043±0.0012) | Koehler et al. (2012) |
| $kdocl$ | Decomposition rate of labile DOC at 20 °C ($day^{-1}$) | 0.14 (0.07–0.14) | Søndergaard and Middelboe (1995) |
| ESD | Equivalent spherical diameter (μm) | 18 | Ruiz et al. (2004) |
| Q10 | Temperature coefficient of organic C decomposition in the water column | 1.5 (1.5–2.0) | Moore and Dalva (2001) |
| pH | pH in the water column | 6.0 (5.8–6.2) | Vachon and Prairie (2013) |
| *Sediment* | | | |



| | | | |
|---|---|---|---|
| $KRCVL$ | Decomposition rate of very labile litter at 20 °C (day$^{-1}$) | 0.25[d] | Gilmour et al. (1985) |
| $KRCL$ | Decomposition rate of labile litter at 20 °C (day$^{-1}$) | 0.074[d] | Gilmour et al. (1985) |
| $KRCR$ | Decomposition rate of resistant litter at 20 °C (day$^{-1}$) | 0.02[d] | Gilmour et al. (1985) |
| $KRB$ | Decomposition rate of labile micro biomass at 20 °C (day$^{-1}$) | 0.12[d] | Molina et al. (1983) |
| $HRB$ | Decomposition rate of resistant micro-biomass at 20 °C (day$^{-1}$) | 0.04[d] | Molina et al. (1983) |
| $KRH$ | Decomposition rate of labile humus at 20 °C (day$^{-1}$) | 0.16[d] | Molina et al. (1983) |
| $HRH$ | Decomposition rate of resistant humus at 20 °C (day$^{-1}$) | 0.006[d] | Molina et al. (1983) |
| $D_{m,DOC}$ | Molecular diffusivity of DOC in sediment pore water ($10^{-9}$ m$^2$ s$^{-1}$) | 0.57 (0.24–1.78 ) | Thoma et al. (1991) |
| $D_{tur}$ | Turbulent diffusion coefficient ($10^{-9}$ m$^2$ s) in the bottom water | 5.0 (1.4–112.7) | Portielje and Lijklema (1999) |
| k | Attenuation of $D_{tur}$ with sediment depth (m$^{-1}$) | 30 (30–125) | Portielje and Lijklema (1999) |

[a] Derived from chlorophyll-*a* concentration using the function developed by Desortová (1981):

[b] Derived from the ratio (4.3–11.4) of DOC: POC collected from boreal rivers.

[c] 1/3 of exudation (10% of GPP) is refractory, and 2/3 of exudation is labile.

[d] Default values in DNDC







**Figure 1**

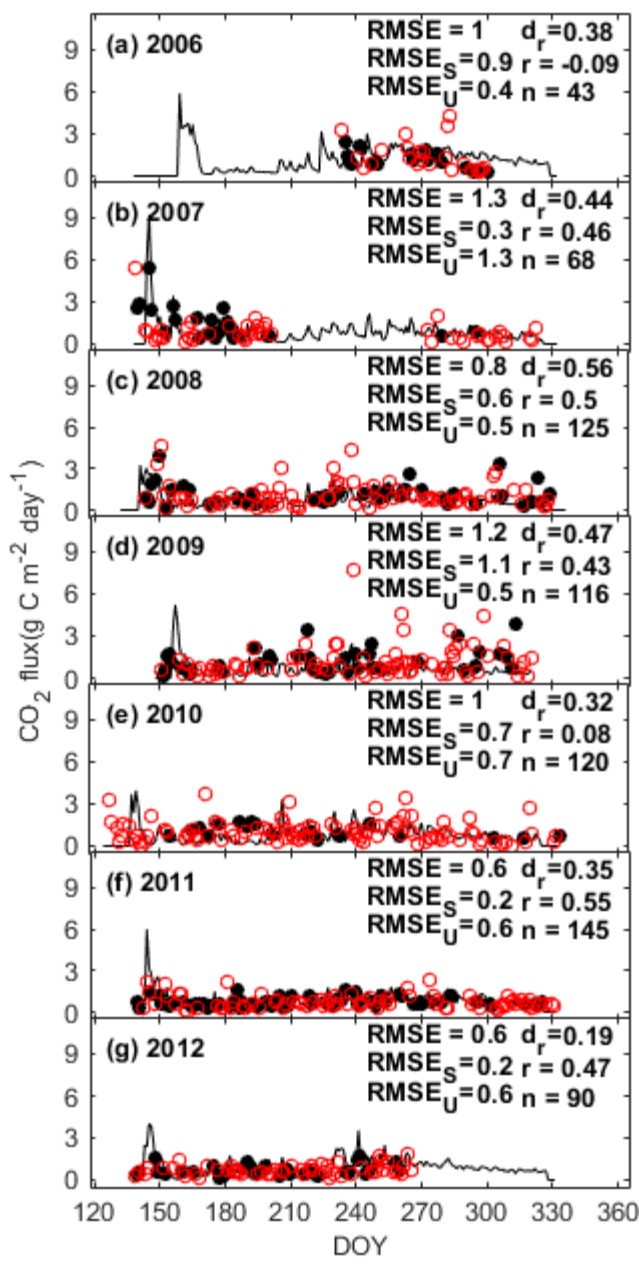

**Figure 2**



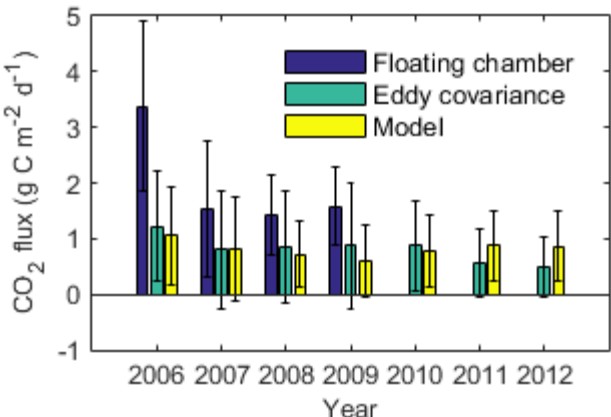

**Figure 3**





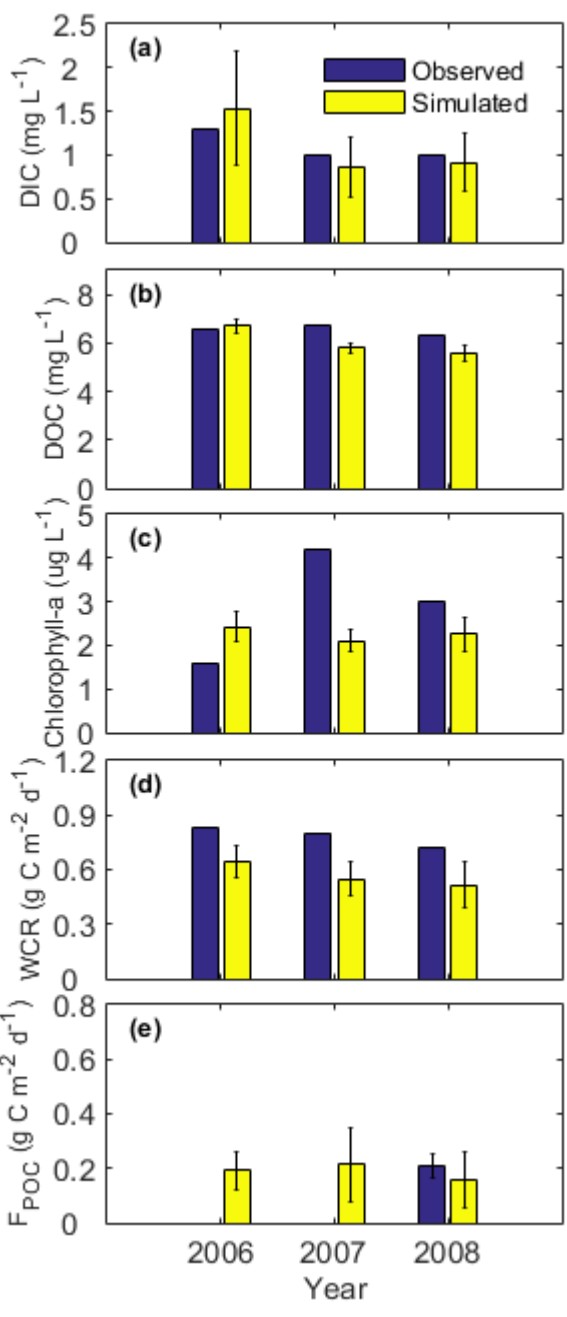

**Figure 4**



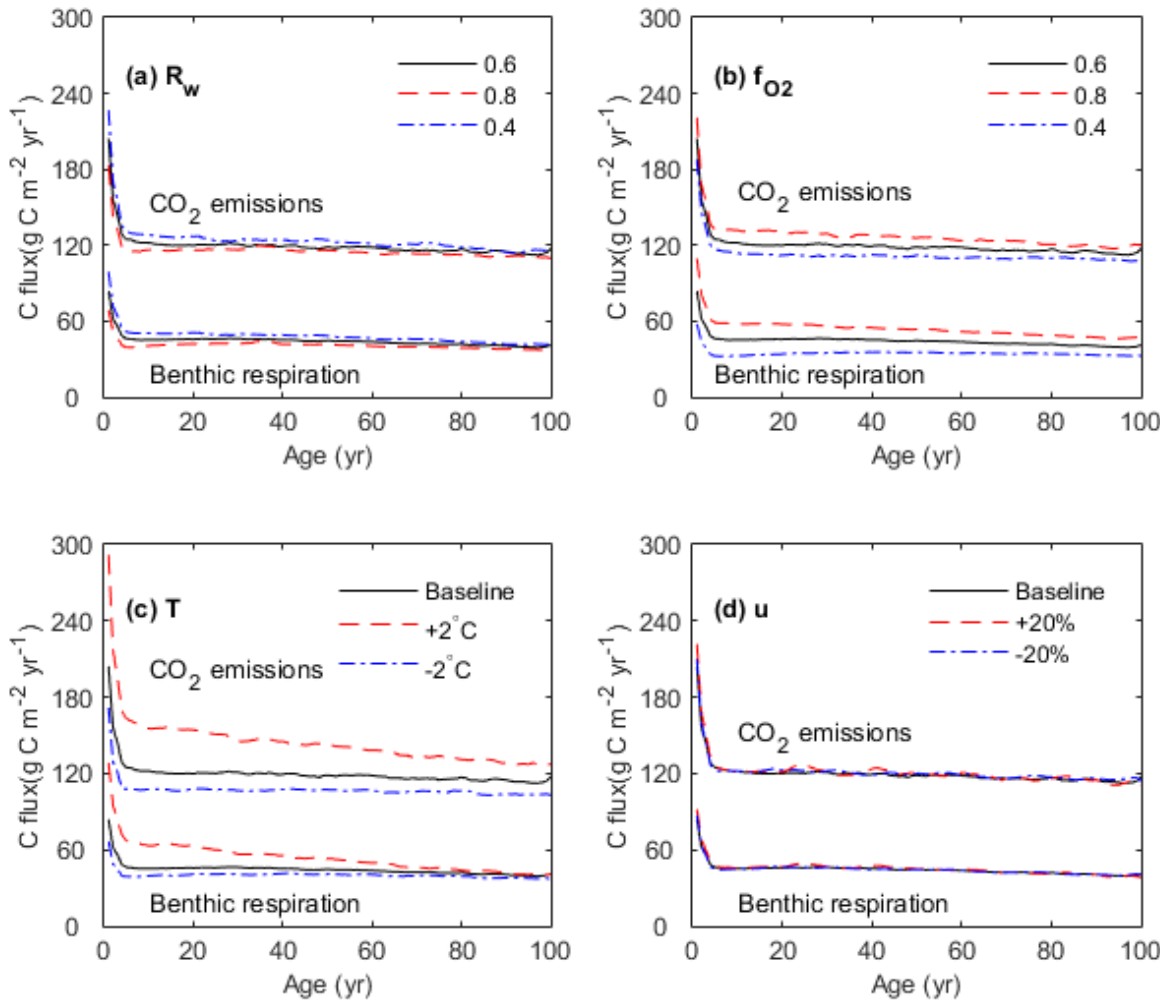

Figure 5



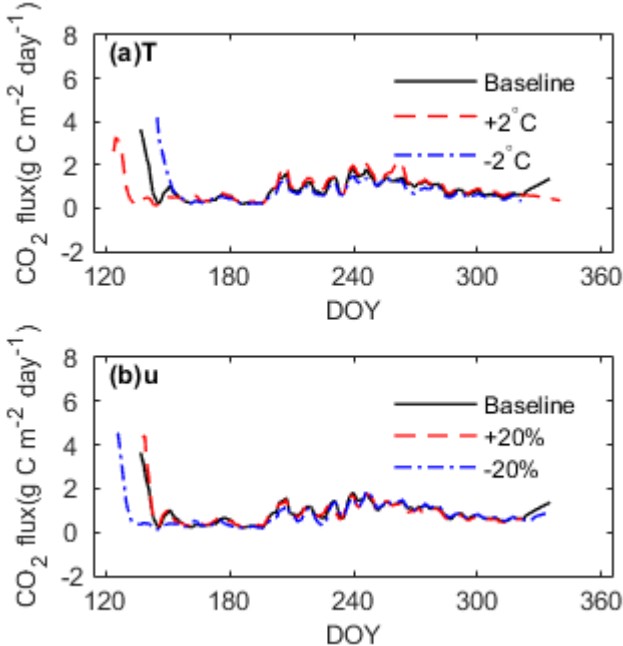

Figure 6





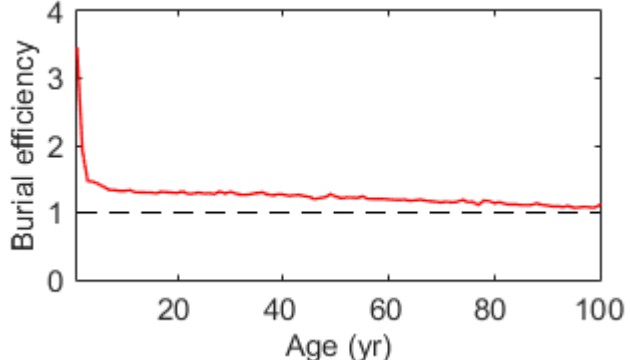

Figure 7

