# Peer review of "Figure S1. Modelled (lines) daily mean water partial pressure of CO2 ( $p\text{CO}_2$ ) and measured (cycles) daily outflow $p\text{CO}_2$ for the Eastmain-1 reservoir from 2007 to 2012."

_Biogeosciences, 2016_

## Referee Comment (RC1) · Anonymous Referee #1 · 7 May 2016

The paper presents a model, developed specifically for calculation of $CO_2$ emissions from hydroelectric reservoirs. To the best of my knowledge of the literature available so far, this is the first mechanistic model of $CO_2$ emissions applied to and validated at a concrete operating reservoir. The model demonstrated generally fair agreement to observations. This work is a substantial step towards process-based modelling assessment of GHG efflux from either existing or planned hydroelectric stations. The perspectives for the development of this work via including methane dynamics and more process-based approaches to simulate C transformations are clear and promising. I have no doubts that the manuscript is worthy to publish in Biogeosciences.

I have a number of specific comments, especially in the model description section.

[Figure]

They are mostly related to justification of model formulations chosen, but also to processes, that are omitted. For instance, the vertical bubble transport of gases and the $CO_2$ flux from turbine flow, are not included, and there is no discussion what it might imply for the model performance. I would also like to see the details of vertical diffusion of DIC between hypolimnion and epilimnion, given there is usually a huge DIC gradient there (BTW, is it the case for Eastmain-1 reservoir?), so that the vertical $CO_2$ flux from bottom waters to surface is controlled by diffusivity coefficient. What is the value for this coefficient used?

The paper lacks comprehensive explanation on the choice of parameters the model sensitivity was studied in respect to. What parameters entering model formulae for biogeochemical processes can be considered as firmly established, and what are loosely defined? Of course, this is a difficult task for such distinction to be made, if even possible so far, but anyway some discussion on this topic should be provided.

All specific comments are added as sticky notes to the original manuscript.

Please also note the supplement to this comment:
http://www.biogeosciences-discuss.net/bg-2016-100/bg-2016-100-RC1-supplement.pdf

**Supplement:**

[revised manuscript text omitted]

Figure 5

[Figure]

[Figure]

[Figure]

Figure 6

[Figure]

[Figure]

[Figure]

Figure 7

---

## Referee Comment (RC2) · Anonymous Referee #2 · 12 May 2016

This paper deals with the modeling of CO2 emissions from the boreal hydroelectric Eastmain-1 reservoir. Eastmain is the most studied boreal reservoir so far in terms of greenhouse gas emissions and therefore the existing database allows the development of process-based models.

GENERAL COMMENTS: The manuscript is topical for the readership of Biogeosciences and such model and its potential development towards methane emissions is of first importance for aquatic biogeochemists working on carbon cycle. The manuscript focuses only on CO2 emissions and could be significantly improved by exploring in details the main processes of the carbon cycle (see major comments), especially when field data exists to compare with the model.

MAJOR COMMENTS:

-The model is a combination of Kim et al., 2016 (Ecological Modeling) and Wang et al, 2016 (Science of the total environment) models. It should be clearly said in the introduction of the model description and more importantly, a comparison with Kim et al. should be given in details since the CO2 emissions are simulated in both papers over 2006-2009 and validated with the same dataset (TBL and EC tower). Is there any improvement with the addition of the water column model? Was the Forest-DNDC model modified compared to the version of Kim et al. 2016?

-The calculations of pCO2 are simplified and it does not take into account carbonate equilibrium. What is the advantage compare to the classical calculations?

-The organization of the section 2 (material and methods) could be improved. It should start with the site description and being followed by a section with a summary of relevant work conducted on this site and used in the publication (in situ measurement and modeling). It is currently spread over the model description, site description, model tests and calibration as list of parameters or values. It would help the reader also to better identify the recent improvement of the model resulting from the addition of the water column. If done, the model description, tests, calibration and validation should be clearer and to the point. The section 2.1.2 is very short and implies to read Wang et al. (2016). A few key equations would be very useful for the reader. The sections 2.3 and 2.4 should be divided in shorter and more focused sections (initial conditions, inputs from watershed, meteorology...).

- In the section 2.1.3, the reader expects a content related to the degradation of allochthonous and/or autochthonous organic matter deposited at the bottom of the reservoir whereas it is mostly about the degradation of the flooded organic matter (soil, vegetation...). This section should probably be divided in two distinct sections to improve the readability. Some sensitivity analysis should be performed on the amount of the flooded organic matter and on the amount of allochthonous DOC/POC.

-The model is a 1-DV model and no vertical profile of modeled variables is shown. Such typical figures are missing to evaluate if how processes are well reproduced by the model or if the model gives "only" a good average value for the "bulk" water column. It would be nice to see data from Teodoru et al (2011) (pelagic and benthic respiration, primary production, benthic respiration) and Demarty et al (2011) (vertical profiles) for instance being used for comparison with the model.

-I would recommended to put the monitoring of the pCO2 in the generation station (supplemental) in the main document since it is the best way to have the average concentration over the whole water column. It also offers the possibility of computing downstream emission.

-A discussion about the pool of carbon fueling emissions would be very interesting: What are the relative contributions of the pelagic respiration, the autochthonous and allochthonous organic matter and the flooded organic matter to the CO2 emissions? Those elements could reinforce the section 4.1 where all sources are listed but no information is given about the main source for the first years and after a few decades..

-the section 4.2 is basically about the sensitivity of the model to temperature change on CO2 emissions. I would be very informative to provide illustrations of temperature change on both the physics (vertical stratification, duration of ice cover...) and on bio-geochemical processes (respiration, PP in the water column, CO2 production in the soils and overlying sediments...). Currently, this section does not provide any quantitative answer to the tested effect.

DETAILED COMMENTS: -P(age)1-L(ine)24: ""engineering" reservoir lifetime (100 years)" could be replace vy the widely-used life-time analysis

-P1-L27: oxygen effects?

-P2-L9-10: Many papers by JJ Cole, Carpenter and their teams or the synthesis by Duarte and Prairie (2005) would be more relevant for the prevalence of heterotrophy in

aquatic ecosystems.

-P2-L11 " water-saturated sediments where the organic matters (e.g., plant biomass, litter, and soil organic matter)": Sediments are different from the flooded organic matter.

-P3L14: what are those "minimum inputs" compare to the listed "sophisticated" models? This should be discussed later on in the manuscript.

-P3-L23-26: "Based on limited empirical data, we test the hypothesis that the boreal reservoir will be a net source of CO2 to the 25 atmosphere. We further hypothesize that the exchanges will be the largest in the first one to two decades and will then show little secular change thereafter—i.e. year-to-year variability around a fairly constant mean" The Eastmain database is not a limited database: 6 years of EC, several field campaigns with floating chamber, DOC, pCO2, respiration, Chloa to cite a few... And the two hypotheses here are not hypothesis since those results are well know (Teodoru et al., 2012). The challenge was rather to check if a simple model is able to reproduce the emissions.

-P4-L21: Is the sentence a title for a section?

-Page 10 Line 20-23: There is no explanation about the tree removal. Was it really done before flooding? If yes, this should be in the site description. Is it a theoretical hypothesis for the evaluation of the role of tree trunk organic matter on emissions and the evaluation of mitigation options?

-P11-L18: what does dr stands for?

-P11-L26-27: This should be extended as noted is the general comments.

-P12-L25-26: "Both increasing and decreasing wind speeds enhanced annual CO2 emissions 25 only by 1 and 1% over 100 years, respectively." Unclear sentence, should be rephrased.

-P12-L29: "grater"... greater

-P13-L17: more information about the pelagic processes is needed since this is where the improvement over Kim et al. (2016) are.

-P13-L20: "Our simulations also show that sediment organic C keeps loosing over the simulation period" needs to be rewritten taking into account that this is very probably the pool of flooded organic matter that loose C instead of the sediment which might accumulate C even if at very low rate.

---

## Short Comment (SC1) · 18 May 2016

I just want to clarify that FAQ-DNDC (this study) is not a combination of FF-DNDC (Kim et al., 2016) and SIWAS (Wang et al., 2016). FF-DNDC and FAQ-DNDC are more like brothers rather than father and son.

Indeed, FF-DNDC is the first attempt to simulate the effects of deep flooding (i.e. reservoir conditions) on soil carbon efflux from forests and peatlands based on the soil biogeochemistry. FAQ-DNDC aims to predict the effects of flooding terrestrial ecosystems (e.g., forests in this study) on reservoir surface CO2 emissions. For both models, the backbone of soil/sediment organic carbon dynamics is "DNDC" (Li et al., 1997). However, the two models are different in terms of the representation of overlying water

column processes due to their different development purposes aforementioned. The FF-DNDC model treats the water column processes using a simple empirical equations and prescribed parameters, such as sedimentation and aquatic primary production rate (0.16 g C m-2 d-1) and the open water period (215 days). These simple functions and prescribed parameters are not used in the FAQ-DNDC model. Instead, to simulate reservoir CO2 emissions, the FAQ-DNDC model includes important physical and biogeochemical processes such as water vertical movement and DOC/DIC/POC dynamics in the water column and gas/mass exchange across air–water and water–sediment interfaces through integrating a lake carbon model (Hanson et al., 2004), a thermal and water stratification model (Wang et al., 2016), and Forest-DNDC (Li et al., 2005, Miehle et al., 2006).

FF-DNDC and FAQ-DNDC are different, although they have similar characters in the sediment organic carbon dynamics. I believe that FF-DNDC should be mentioned in the introduction other than in the model description.

Q: "Is there any improvement with the addition of the water column model?"

A: In this study, the lake carbon model (Hanson et al., 2004) and the thermal and water stratification model (Wang et al., 2016) are integrated to simulate carbon processes in the water column, which is the "future plan" for FF-DNDC to predict reservoir surface CO2 emissions (Kim et al., 2016). Please note that FF-DNDC only includes a simple empirical-based water column module that does not include detailed biogeochemical processes like DOC decomposition in the water column in terms of its purpose. However, in the FAQ-DNDC model, the biogeochemical and physical processes such as DOC, POC, DIC dynamics in the water column, gas exchange across air–water interface, and mass transfer across water–sediment interface are considered (see section 2.1.1).

Q: "Was the Forest-DNDC model modified compared to the version of Kim et al. 2016"

A: There are many differences in modifying Forest-DNDC between FF-DNDC and FAQ-

**BGD**

DNDC. FF-DNDC modified the parameters and functions of DNDC for the flooded simulations. However, the most modifications (e.g., new DIC pool in each sediment layer, diffusion processes within the sediment and sediment-water interface) described in section 2.1.3 in this study are not applicable for FF-DNDC.

Reference:

Hanson PC, Pollard AI, Bade DL, Predick K, Carpenter SR, Foley JA (2004) A model of carbon evasion and sedimentation in temperate lakes. Global Change Biology, 10, 1285-1298.

Kim Y, Roulet NT, Li C et al. (2016) Simulating carbon dioxide exchange in boreal ecosystems flooded by reservoirs. Ecological Modelling, 327, 1-17.

Li C, Frolking S, Crocker GJ, Grace PR, Klír J, Körchens M, Poulton PR (1997) Simulating trends in soil organic carbon in long-term experiments using the DNDC model. Geoderma, 81, 45-60.

Li C, Trettin C, Sun G, Mcnulty S, Butterbach-Bahl K (2005) Modeling carbon and nitrogen biogeochemistry in forest ecosystems. In: 3rd International Nitrogen Conference. (eds Zhu Z, Minami K, Guangxi X) pp Page, China, Nanjing, Science Press.

Miehle P, Livesley SJ, Feikema PM, Li C, Arndt SK (2006) Assessing productivity and carbon sequestration capacity of Eucalyptus globulus plantations using the process model Forest-DNDC: Calibration and validation. Ecological Modelling, 192, 83-94.

Wang W, Roulet NT, Strachan IB, Tremblay A (2016) Modeling surface energy fluxes and thermal dynamics of a seasonally ice-covered hydroelectric reservoir. Science of the Total Environment, 550, 793-805.

---

## Referee Comment (RC3) · Anonymous Referee #3 · 10 Jun 2016

The authors represent a modelling approach to quantify CO2 emissions by integrating a terrestrial and an aquatic model. They applied their model framework to assess the effects of reservoir creation and the following CO2 emission on carbon dynamics. In my opinion this work represents an important step towards a more complete understanding of the carbon cycle. It stresses the importance to integrate terrestrial and aquatic systems to fully map the different components of the carbon cycle, which is especially importance with respect to climate change. Although I see several issues that should be solved first, I recommend publishing the manuscript in Biogeosciences after revision.

Major comments

(A) One concern is that to me it seems that the authors did the calibration of their model and the validation not with independent data. It should be made more clear which data have been used for calibration (P1 L16 'using the measurements ... in a ~600 km2 boreal hydroelectric reservoir, Eastmain-1') and for validation ('We then evaluated the model performance against observed $CO_2$ fluxes data from an eddy covariance tower in the middle of the EM-1 reservoir').

(B) For comparison of observed and simulated data, or rather expected and simulated data I'd like to see more statistical tests to show if the differences are significant or not. Statements as 'reasonably well' (P1, L22) or 'greatly influence' (P12, L28) are not sufficient.

(C) The conducted sensitivity analysis only includes Rw, FO2, air temperature and wind speed. It would have been interesting to see how sensitive the model reacts on the most important parameters controlling the processes in the water column, like decomposition rates of POC and DOC.

(D) It did not appear clear to me, how much of the model developments has been originally done by the authors and how much of their framework relies on the work of others. This should be stated clearly and possibly also indicated in an overview figure as Figure 1.

(E) I see some potential for improvement in the discussion.

- P 13 LL14: The empirical model shows a decline period of 12 to 15 years. Another study (not including water column processes) estimates the period to be several decades. The author's model (including water column processes) estimated a period of only 3 years. It should be more clearly discussed that this inconsistency (a) shows the importance of including the water column processes and (b) shows that the implementation in the model can still be improved.

- The authors state (P13 L25) that they 'did not incorporate methane production'.

Please explain why do you think that 'if methane production is included, the reservoir would probably need more time than reported here'.

Minor comments

(A) It would be helpful to show the sequence of processes calculated in the model. I assume this could be added to Figure 1.

(B) Some specific questions: Why is GPP depending on DOC (Equ.1)? How did the authors convert form biomass to carbon (P5 LL5)?

(C) Figures:

- The figures only partly support the message of the manuscript (e.g. Figure 4 and Figure 5).

- Figure 1 is not clear enough to show the important processes and fluxes in the model. Some arrows seem to come from nowhere (e.g. the solid line arrow 'Incident solar radiation', or one solid black line going down from the 'Passive humus pool'). Please clarify.

- In Figure 2 one cannot distinguish the temporal patterns the authors refer to in the text, especially in (c), (d) and (e).

- Figure 3. The model's reaction seems relatively constant over the years, which could be a sign that important processes might be ignored or their implementation should be improved. Please discuss.

- Figure 4. I can only partly agree with the conclusions the authors draw from the data shown in this figure, especially for the sinking rate which seems to be poorly reproduced. Why do the observed data don't have any variance?

- Figure 5. The figure does not support the statement that the 'annual CO2 emissions and benthic dissolved CO2 fluxes showed great sensitivity to Rw'. There seems to be some sensitivity but I would not call it 'great'. And besides I'd like to see more statistics

on the significance of differences in general.

- Figure 7. What does this figure show? It doesn't fit to the caption.

- Figure captions should include more units. E.g. caption for Figure 5: 'The curves were smoothed using a moving average filter with a span of 5.' I assume 5 years. And in the caption for Figure 6 'The curves were smoothed using a moving average filter with a span of 5.' Here I assume days.

- I would rather keep Figure S1 in the main part and remove others (e.g. Figure 2 and Figure 7) to the supplementary.

(D) Tables:

- Table 1. 'Thickness of mineral soil' is assumed to be 80cm. Please indicate/explain the basis of this assumption? Especially because it strongly influences the soil organic C in mineral soil and therewith the size of this carbon pool. What are the numbers in parenthesis for 'soil organic C (based on Paré et al 2011)?

- Table 2: On which data are the assumptions based?

(E) There are some language issues and typos in the manuscript (e.g. caption Figure 6 'celerity' should be 'clarity'; P2 L29 'in a seasonally ice-covered lakes'; P9 L5 'can generates').

---

## Author Comment (AC1) · 5 Jul 2016

Reviewer #1

We appreciated for the comments and suggestions that significantly improve the quality of the manuscript. We have addressed referee 1's comments point by point and will make changes in the revised manuscript, which are detailed below.

1.  The paper presents a model, developed specifically for calculation of CO2 emissions from hydroelectric reservoirs. To the best of my knowledge of the literature available so far, this is the first mechanistic model of CO2 emissions applied to and validated at a concrete operating reservoir. The model demonstrated generally fair agreement to observations. This work is a substantial step towards process-based modelling assessment of GHG efflux from either existing or planned hydroelectric stations. The perspectives for the development of this work via including methane dynamics and more process-based approaches to simulate C transformations are clear and promising. I have no doubts that the manuscript is worthy to publish in Biogeosciences.

    *Author response: We thank the referee 1 for his/her positive overall evaluation of our study.*

2.  I have a number of specific comments, especially in the model description section. They are mostly related to justification of model formulations chosen, but also to processes, that are omitted. For instance, the vertical bubble transport of gases and the CO2 flux from turbine flow, are not included, and there is no discussion what it might imply for the model performance. I would also like to see the details of vertical diffusion of DIC between hypolimnion and epilimnion, given there is usually a huge DIC gradient there (BTW, is it the case for Eastmain-1 reservoir?), so that the vertical CO2 flux from bottom waters to surface is controlled by diffusivity coefficient. What is the value for this coefficient used?

    *Author response: We appreciate reviewer 1's comments and suggestions on model formulation. Yes, we tested the model using data collected from the Eastmain-1 reservoir.*

    *The bubble emission pathway is important and will be considered in our next manuscript that focuses on methane and oxygen dynamics in the water column and sediment. The bubbles typically are composed of gases of $CO_2$, $CH_4$, and $N_2$. As the $CO_2$ production in the sediment may not be increased, the lack of bubble emission pathway will not have significant impacts on total $CO_2$ emissions.*

    *Degassing from the turbine flow is beyond the scope of the current study, but discussion on the issue is provided in the revised manuscript. Our partners in this research, Hydro-Quebec, measure the gas concentration from the outflow of the turbines so there are good empirical observations. One could, in principle, develop an empirical relation between flow through the powerhouse and gas emitted, but the actual flows are difficult to obtain because they are proprietary information as it can influence the price of electricity.*

    *In the model, vertical exchange of DIC between hypolimnion and epilimnion is controlled by water mixing (convection), as diffusion is very inefficient. For example, the epilimnion deepening during the summer bring DIC from hypolimion to the epiliminion. We added*

*sentences to describe how the model mix the water and its solute (e.g., DOC). We published a paper this year that describes the convective mixing in this model (Wang et al. 2016).*

*Reference:*

*Wang W, Roulet NT, Strachan IB, Tremblay A. Modeling surface energy fluxes and thermal dynamics of a seasonally ice-covered hydroelectric reservoir. Sci. Total Environ.550: 793-805, 2016.*

3. The paper lacks comprehensive explanation on the choice of parameters the model sensitivity was studied in respect to. What parameters entering model formulae for biogeochemical processes can be considered as firmly established, and what are loosely defined? Of course, this is a difficult task for such distinction to be made, if even possible so far, but anyway some discussion on this topic should be provided.

   *Author response: The information about why only chose these four parameters has been added in the revised manuscript. The similar issue has been addressed by the other two reviewers (see R2C5, R2C9 and R3C4; R: reviewer, C: comment) in terms of different perspectives.*

   *Briefly speaking, we are interested to know how flooding terrestrial organic carbon influence post-flooded reservoir $CO_2$ emissions, which is the main purpose of this model and this study. The sensitivity analysis for the parameter of aboveground biomass removal shows the amount of flooded organic carbon significantly and positively influence $CO_2$ emissions. For the oxygen effect parameter (in the revised manuscript, we changed it to "partitioning coefficient of decomposition production"), we interested to understand if the lack of oxygen cycle in the model significantly affected the simulated emissions. The results indicate that incorporating oxygen cycle would improve the quality of the output and we are now in the process of adding this to the model as we add in methane. Further, we want to know how and by what mechanisms the environmental factors (air temperature and wind speed) influenced the carbon emissions using the process-based model, as these two climate variables have more significant influence on thermal dynamics and carbon emissions.*

   *We have now categorized all physical and biological parameters (Table 1) as hardware parameters. Model sensitivity for most existing parameters in their original model have been investigated in previous studies (e.g., Zhang et al., 2002).*

   *Reference:*

   *Zhang, Y., Li, C., Trettin, C. C., Li, H., and Sun, G.: An integrated model of soil, hydrology, and vegetation for carbon dynamics in wetland ecosystems, Global Biogeochem. Cycles, 16, 1061, 10.1029/2001GB001838, 2002.*

Minor comments:

4. P1L27: what does it mean "positively enhance"? Did you mean simply "enhance"?

*Author response: We have deleted "positively".*

5. P1L28: seeming a contradiction: isn't CO_2 flux the same as CO_2 emission?

   *Author response: $CO_2$ flux across air-water interface is the same as $CO_2$ emission. We rephrased the sentence to avoid the confusion.*

6. P1L30–31: do you mean, larger wind speed makes open water period shorter? Please, make it clearer

   *Author response: Yes. Higher wind speed leads to larger heat loss (higher latent and sensible heat fluxes to the atmosphere), resulting in a shorter open water period (Wang et al., 2016). We rephrased the sentence to make it clearer.*

   *Reference:*
   *Wang W, Roulet NT, Strachan IB, Tremblay A. Modeling surface energy fluxes and thermal dynamics of a seasonally ice-covered hydroelectric reservoir. Sci. Total Environ.550: 793-805, 2016.*

7. P2L20: …provide…with…

   *Author response: Did as suggested.*

8. P2L27: air temperature? Wind speed.

   *Author response: Yes, air temperature. We specified it in the revision.*

9. P2L27: Wind speed.

   *Author response: Did as suggested.*

10. P3L22: assess

    *Author response: Did as suggested.*

11. P3L31 climate data inputs

    *Author response: Did as suggested.*

12. P4L2: I would say: verified, calibrated

    *Author response: Did as suggested.*

13. P4L3: simulates

    *Author response: Did as suggested.*

14. P4L5: temperature

   *Author response: Did as suggested.*

15. P4L7: not all of them, as methane dynamics is not included

   *Author response: Yes, you are right. None of models can simulate all processes. DNDC actually is able to produce methane in the sediment through simulating soil redox chemistry. We are in the process of attempting to develop a methane module for FAQ-DNDC. Please also see our response to comment 2, R2C1, and R3C7 about the methane module and its potential impacts on the current study.*

16. P4L8: I would say: … are implemented as follows.

   *Author response: Did as suggested.*

17. P4L13: … are simulate:…

   *Author response: Did as suggested.*

18. P4L18: Please, precise, how do you mix between epilimnion and hypolimnion using SIWAS.

   *Author response: We added the description about water and its solute mixing in section 2.1.2 Thermal dynamic and water mixing. Please see our response to the comment 2.*

19. P4L21: The sentence seems incomplete

   *Author response: We rephrased the sentence.*

20. P4L23: How do you specify DIC and DOC concentration in atmospheric precipitation? Are there observations data on this concentration?

   *Author response: For parameterizing these two carbon input variables from the atmospheric precipitation, we conducted literature search and consulted experts (Dr. Tim Moore, McGill University) to select 0.6 and 2.0 mg/L for DIC and DOC, respectively.*

21. P4L24: However, snow is made up by atmospheric precipitation, which includes DOC and DIC. Have you estimated, how much of annual atmospheric DOC&DIC input is contained in solid precipitation?

   *Author response: We do not estimate the amount of atmospheric DOC or DIC input stored in solid precipitation (snow). To our best knowledge, few studies focus on DOC/DIC concentration in solid precipitation. We think the atmospheric deposition inputs (either DOC or DIC) has insignificant effects on total carbon dynamics compared to inputs of carbon from inflow in our study reservoir.*

22. P4L31-32: I would replace all these numeric values by symbols, and provide a table with values standing behind these symbols. Thus, you underline that these parameters should be, strictly speaking, reservoir-dependent, and not constants like physical constants. This also applies to other biogeochemical parameters used in your paper below, so I do not return to this question there.

**Author response:** We did as suggested. The table would be like below:

| Equations | $b_0$ | $b_1$ | $b_2$ | $b_3$ | $b_4$ |
|---|---|---|---|---|---|
| 1 GPP | 0.80 | −0.67 | 0.75 | 1.33 | −0.77 |
| 1 PR | 0.67 | −0.94 | 0.77 | 1.28 | −0.64 |
| 3 | 1.58 | 4.97 | - | - | - |
| 6 | −0.453 | 0.71 | −0.087 | - | - |
| 11 | 2.51 | 1.48 | 0.39 | - | - |
| 12 | 1911.1 | -118.11 | 3.4527 | 0.4132 | - |
| 16 | -17.0 | 0.06 | - | - | - |

23. P4L31-32: I don't understand how do you avoid using atmospheric radiation fluxes in calculating GPP. I understand that you try to minimize the input atmospheric data and cite the other work in eqs. (1) and (2), but would you please provide a comment, how is radiation regime is implicitly included here (as it should be).

*Author response: We estimated GPP and PR using regression models other than biogeochemical equations. The FAQ-DNDC model itself calculates radiation and heat fluxes. We will investigate if biogeochemical photosynthesis models are better to be used for aquatic photosynthesis in the future but we have to keep the input requirements to a minimum since data is extremely scarce for the northern boreal regions we are interested in.*

24. P5L1 : Please, specify, what are the definitions tou use for mixing depth and sunlight depth.

*Author response: We revised the manuscript to have the detailed description of these two variables. Mixing depth indicates the depth of epiliminon, while the sunlight depth is the depth of the water that is exposed to certain intensity ($>0.03$ W/$m^2$) for irradiation in one layer) of sunlight.*

25. P5L4 : Not clear, is this ratio is assumed in the inflows of a reservoir, or it is a fixed ratio inside a reservoir?

*Author response: This fixed ratio is only for exudation of GPP (like root exudation of terrestrial plants). We modified the sentence to make it clearer.*

26. P5L13: In physical equations I recommmend to avoid using multi-latter notations, like ESD. Consider replacing by single-letter symbol, like D (for diameter).

*Author response: Did as suggested. We changed multi-letter notation to single-letter symbol for non-common variables.*

27. P5L20: What is the value for Q10?

   *Author response: Q10 = 1.5 listed in the table 2. We added its value in the text and removed it from the table.*

28. L5P23: I doubt if it is correct to call it Fick's law, as it there is not a gradient of concentration, but a difference of concentration across a phase boundary. Please, check.

   *Author response: We rephrased our sentence to avoid this confusion.*

29. L5P24 There is one more emission pathway for $CO_2$ in resevroirs -- that is through turbines. If the turbines are located deep, they extract water from hypolimnion with much higher $CO_2$ concentration, compared to epilimnion. Do you take it in to account? Or you can provide estimates arguing it is insignificant for that particular reservoir you simulate in the paper?

   *Author response: Degassing is an important pathway of greenhouse gases for hydroelectric reservoirs. Unfortunately, we do not have direct measurement. So, we do not take it into account in this study yet. The carbon budget estimation using the process-based model is interesting, but we need to consider the effects of flooding on different pre-flooded landscapes (forests, peatlands, and lakes) to giving a whole picture. See our response 2.*

30. P5L25: better to call it "piston velocity", as "usual" diffusion coefficient is measured in m**2/s

   *Author response: Did as suggested. We revised our expression for equations 7-9.*

31. P5L25: what is less than $z_{mix}$?

   *Author response: Considering the daily time-step, $CO_2$ diffusion coefficient in number should be less than mixing depth.*

32. P5L26: Solubility

   *Author response: Did as suggested. We simplified the equations about air–water gas exchange in terms of Henry's law in the revised manuscript.*

33. P6L2: please make the brackets higher

   *Author response: Did as suggested.*

34. P6L4: it is a reference piston velocity at Schmidt number = 600.

   *Author response: We revised our related text.*

35. P6L9: please, find a single-latter notation

   *Author response: Did as suggested.*

36. P6L14: As this is a physical formula, I would prefer if you provide a combination of physical constants providing this value.

   *Author response: The number of 83333.3 is for unit conversion other than physical constants. We listed it as unit conversion coefficient in the revised manuscript.*

37. P6L23: what is a difference between transmission and diffusion?

   *Author response: We revised it to "light transmission and heat transfer".*

38. P7L5: I wonder, what is the minimal vertical diffusion coefficient you use in the model. During stratified periods, metalimnion (thermocline) is almost laminar in lakes and reservoirs, and vertical diffusion is almost molecular, and hence very inefficient. But, given a typically huge concentration gradient of $CO_2$ in metalimnion, the upward $CO_2$ flux should be very sensitive there to diffusivity.

   *Author response: In the model, vertical diffusion of DIC/DOC between hypolimnion and epilimnion is controlled by water mixing, as diffusion is very inefficient. For example, the epilimnion deepening during the summer bring DIC from hypolimion to the epiliminion. We added sentences to describe how the model mix the water and its solute (e.g., DOC). See our response to the comment 2.*

39. P7L6: What do you mean by stirring here? TKE equation includes only production by buoyancy and shear.

   *Author response: Stirring is induced by wind, which is more important than shearing. The surface mixing algorithm follows the DYRESM model (Imberger and Patterson, 1981).*

40. P7L14: Exist

   *Author response: Did as suggested.*

41. P7L24: is that correct that woods are immediately added to litter? What is the typical time for stems fall onto sediments? what do you mean by this: "... water depth is equal to mean water depth"?

   *Author response: It is might not quite right to have all living biomass immediately added. There is not a lot known about plant mortality in flooded and partially flooded conditions. There is no question that wood would decompose slowly. The wood is added to litter but the slowly decomposing pool. In the model, we did not assume that terrestrial plants die immediately, while there is a time lag by assuming that trees die when reservoir water depth*

*reaches the mean water depth. We simulated the water filling process (from November 2005 to June 2006). A more difficult issue to deal with is how the main woody parts of trees is dealt with physical when a reservoir is created. If the trees are frozen into the ice cover the volume of reservoir can be managed to mechanical remove the trees. The logs then float to the surface and can be salvaged over the next few years before they sink. However, in his form of FAQ-DNDC we do treat the wood as recalcitrant litter and if wood were removed it would have to be treated as an additional loss.*

42. P8L3: I would not start sentences with abbreviations or notations

*Author response: We revised the sentence to avoid this.*

43. P8L4: I expect high CH_4 concentrations there leading to bubble formation, so that bubbles transport CH_4 and CO_2 directly to the atmosphere. Do you think this pathway for CO_2 from sediments is insignificant?

*Author response: Yes, bubbles may contain $CH_4$ and $CO_2$. In this study, we do not separate the emissions pathways as we are developing the methane dynamic sub-model (oxidation in the water column and emission pathways). We think the lack of this pathway for $CO_2$ may not significantly influence the total amount of $CO_2$ emissions through the water surface. The CO2 production in the sediment does not change. Also see our response to the comment 2.*

44. P8L7: d^2 C/dz^2

*Author response: Thank you for correcting.*

45.  P8L9: "diffusion coefficient" can't be "a sum of processes". Please rephrase

*Author response: Did as suggested.*

46. P8L12: What is the nature of "turbulence" expected here in porous soil?

*Author response: To our knowledge, in porous sediment, the bottom reservoir water may influence the top porewater. In the thermal module, this has been neglected as the influence is slight compared to the whole water column. However, for the carbon cycle, the turbulent influence cannot be neglected. We used $D_{tur}$ to simulate the effects of water flow on porewater carbon diffusion. .*

47. **P8L16: i-th**

*Author response: Did as suggested.*

48. P8L25: I would expect this coefficient representing not only effective diffusion on the soil side, but also at the water side

*Author response: This diffusion coefficient only works on the diffusion across water-sediment interface. On either soil or water side, they have different diffusion coefficient.*

49. P9L5: megawatt-hours units are used to quantify the total energy produced for a specified period, 1 day, 1 yr, etc. If this period is not specified, they use megawatt.

   *Author response: We indicate the installed capacity. We re-wrote the sentence.*

50. P9L7: OK, does it mean that roughly 32% of annual atmospheric DIC and DOC are gained by reservoir in form of snow?

   *Author response: Here just shows the climate in our study region so that there is no any implications for the carbon cycle. We assume that no DOC or DIC exists in snow in this study.*

51. P9L20: As the eddy covariance system was deployed at the island, are there estimates how much of measured EC $CO_2$ flux originated from the island?

   *Author response: The reservoirs fluxes are only for when the towers 'sees' the water surface. The land sectors from the island were not included in our analysis when they were in the footprint. We have added a sentence to the methods to make this clear.*

52. P9L26: was the depth of water uptake by generation station taken into account, to compare $CO_2$ measured and simulated?

   *Author response: Unfortunately, we do not have such information about the depth of intake hole in the dam. We assume that the outflow represents the mean water conditions, as FAQ-DNDC is a one-dimensional model.*

53. P9L31 : in river?

   *Author response: yes, in river. This is calculated for river POC input.*

54. P10 L29 two?

   *Author response: Actually, we used three methods: root mean square error (RMSE), refined Willmott index (dr), and Pearson corrleation coefficient (r).*

55. P11L3: I would expect here a rationale, why only these two parameters from a large number of biogeochemical parameters used in the model, were selected for sensitivity runs.

   *Author response: We have revised the paragraph to explain why these four parameters were selected. Please also see our response to the comment 3.*

56. P11L13: I wonder, if you could make up a budget of $CO_2$ in the reservoir, based on your model? I.e. calculate the contribution of all internal/external sources and sinks of $CO_2$ into $CO_2$ emission?

*Author response: Definitely yes, the model can be used to estimate the carbon budget of the reservoir. However, the current study focuses on model development, calibration, and testing. Because of the spatial heterogeneity of CO2 emissions, a completed carbon budget has to include the effects of flooding on different pre-flooded landscapes. This would make the model development paper lose its focus. However, a carbon budget is the subject of a subsequent paper.*

57. P11L18 : What is d_r?

    *Author response: d_r is the revised Willmott index. We added the notation in the methods part in the revised mansucript. This is also been mentioned by reviewer 2.*

58. P12L13: significant?

    *Author response: yes, it means significant. We revised "great" to "significant".*

59.  I couldn't find in the model description section, was terrestrially-derived DOC , DIC and POC attributed to inflows only or groundwater discharge was explicitly taken into account as well?

    *Author response: Yes, terrestrially-derived DOC, DIC, and POC attributed to inflows, whereas groundwater discharge was not taken into account. The reservoirs we study in northern Quebec are located on the Canadian Shield. It is the topography created in this landscape along with the generally low permeability of the igneous rock of the Shield that makes the region suitable for reservoir creation. Groundwater could be an important component in other geographical regions but we have not included it in our study because it is not an issue.*

60. P13L13 : declines

    *Author response: Did as suggested.*

61. P13L25-26: You don't include the possible role of CO_2 ebullition

    *Author response: Bubbles will be considered in our next manuscript, but we think it will not significantly increase $CO_2$ emissions. See our response to comments 43 and 2.*

62. P14L16 Occur

    *Author response: Did as suggested.*

63. P14L31: what do you mean by water vertical movement here? Is it convection, that is directly affected by thermal dynamics?

    *Author response: Water vertical movement indicates spring/autumn turnover, summer stratification. We rephrased the sentence to avoid the confusion.*

64. P15L16: I don't see a link between summer CO_2 flux and diurnal variation. The reader could look into cited literature, of course, but could you be more clear here, please?

   *Author response: Here we were arguing the second reason (high GPP in summer due to high water temperature) for lower summer $CO_2$ fluxes. Higher GPP at daytime than at nighttime leads to lower $CO_2$ emissions. We rephrased the sentence to make our point clearer.*

65. P15L19: what does it mean: dissolved CO_2 ... can be filled up ...?

   *Author response: Here we wanted to say that dissolved $CO_2$ (a component of DIC) in the epilimion increased with epilimion deepening (mixing the upper layer of hypolimion). We re-wrote the sentence*

66. P16L2: What aspects? Please precise

   *Author response: Basically, we do not consider carbonate equilibria in the current model formulation. We rephrased the sentence.*

67. P16L17: Typo?

   *Author response: should be "remaining".*

68. Figure 1, Are there radiation fluxes?

   *Author response:FAQ-DNDC does not require radiation fluxes as inputs, but calculates the radiation fluxes (short-wave and long-wave radiation, heat fluxes) by its inner algorithms based on latitude and an average cloud cover parameter. Since these inputs variables are seldom available as standard observations we continue to use DNDC's computed radiation fluxes. This approach has its advantages for general use but is a disadvantage if actual observations were available.*

69. I suggest to rearrange this figure: 1) make two columns of plots 2) increase the vertical scales, as now both measured and simulated time series are at the bottom of plots and hardly discernable

   *Author response:Did as suggested. We also changed the symbols to open triangles for the clarity. This is also addressed by reviewer 3.*

---

## Author Comment (AC2) · 5 Jul 2016

Reviewer #2

We appreciated for the comments and suggestions that significantly improve the quality of the manuscript. We have addressed referee 2's comments point by point and will make changes in the revised manuscript, which are detailed below.

1. This paper deals with the modeling of CO2 emissions from the boreal hydroelectric Eastmain-1 reservoir. Eastmain is the most studied boreal reservoir so far in terms of greenhouse gas emissions and therefore the existing database allows the development of process-based models. The manuscript is topical for the readership of Biogeosciences and such model and its potential development towards methane emissions is of first importance for aquatic biogeochemists working on carbon cycle. The manuscript focuses only on CO2 emissions and could be significantly improved by exploring in details the main processes of the carbon cycle (see major comments), especially when field data exists to compare with the model.

   *Author response: Yes, this manuscript only focuses on $CO_2$ emissions. We simulated reservoir surface $CO_2$ emissions for the former mature forest area where most observational data (including fluxes measured from EC tower and floating chamber) are available. This manuscript does not intend to estimate the whole carbon budget for the boreal reservoir. We do in the future plan to the model to include methane but the description of the core carbon is already more than enough material for a single manuscript.*

MAJOR COMMENTS:

2. The model is a combination of Kim et al., 2016 (Ecological Modeling) and Wang et al, 2016 (Science of the total environment) models. It should be clearly said in the introduction of the model description and more importantly, a comparison with Kim et al. should be given in details since the CO2 emissions are simulated in both papers over 2006-2009 and validated with the same dataset (TBL and EC tower). Is there any improvement with the addition of the water column model? Was the Forest-DNDC model modified compared to the version of Kim et al. 2016?

   *Author response: FAQ-DNDC includes a water column carbon sub-model and thermal dynamic and water stratification sub-model. It replaces many prescribed parameters used in FF-DNDC. Compared to FF-DNDC, the new model, FAQ-DNDC, has a relatively complete lake carbon cycle including DOC, DIC, POC dynamics. There are many improvements over Kim e al 2016. For example, one of the problems identified in Kim et al (2016) was the sensitivity of the modelled emissions to sedimentation of new production. In Kim et al. (2016) the sedimentation of new production was not known, hence the different guesses at this value. In FAQ-DNDC there is no need to make this guess since new production in now estimated. Modifications for Forest-DNDC in FAQ-DNDC described in section 2.1.3 are different with FF-DNDC (Kim et al., 2016).*

   *Reference:*

*Kim, Y., Roulet, N. T., Li, C., Frolking, S., Strachan, I. B., Peng, C., Teodoru, C. R., Prairie, Y. T., and Tremblay, A.: Simulating carbon dioxide exchange in boreal ecosystems flooded by reservoirs, Ecol. Model., 2016, 327, 1-17*

3. The calculations of pCO2 are simplified and it does not take into account carbonate equilibrium. What is the advantage compare to the classical calculations?

   *Author response: We used DIC-pH to calculate pCO2. This approach could be more accurate compared to the pH-alkalinity approach, as pCO2 might be overestimated from pH and alkalinity in acidic freshwaters (Abril et al., 2015). In the case of EM-1 the underlying geology is igneous rock so source of carbonate is very low. This would be something that needs to be considered in sedimentary catchments with more buffered systems.*

   *Reference:*

   *Abril G, Bouillon S, Darchambeau F, Teodoru CR, Marwick TR, Tamooh F, et al. Technical Note: Large overestimation of pCO2 calculated from pH and alkalinity in acidic, organic-rich freshwaters. Biogeosci. 2015; 12: 67-78*

4. The organization of the section 2 (material and methods) could be improved. It should start with the site description and being followed by a section with a summary of relevant work conducted on this site and used in the publication (in situ measurement and modeling). It is currently spread over the model description, site description, model tests and calibration as list of parameters or values. It would help the reader also to better identify the recent improvement of the model resulting from the addition of the water column. If done, the model description, tests, calibration and validation should be clearer and to the point. The section 2.1.2 is very short and implies to read Wang et al. (2016). A few key equations would be very useful for the reader. The sections 2.3 and 2.4 should be divided in shorter and more focused sections (initial conditions, inputs from watershed, meteorology…).

   *Author response: Agree. We restructured this section as proposed by the reviewer.*

5. In the section 2.1.3, the reader expects a content related to the degradation of allochthonous and/or autochthonous organic matter deposited at the bottom of the reservoir whereas it is mostly about the degradation of the flooded organic matter (soil, vegetation: : :). This section should probably be divided in two distinct sections to improve the readability. Some sensitivity analysis should be performed on the amount of the flooded organic matter and on the amount of allochthonous DOC/POC.

   *Author response: We do recognize that it is an interesting question about the contribution of allochthonous and autochthonous organic matter to $CO_2$ emissions in aquatic ecosystems. However, the environmental problem that we are facing is huge amounts of organic carbon (including soils and vegetation biomass) were buried due to the creation of the reservoir. Therefore, this study focuses on when and how this buried carbon emits to the atmosphere. Secondly, we do not distinguish autochthonous organic matter (POC) deposited at the bottom of the reservoir and flooded soil organic matter or sediment organic carbon. Like*

*littering in the terrestrial ecosystem, POC deposited to the sediment directly entering into sediment carbon dynamics.*

*The sensitivity analysis on the amount of the flooded organic matter has been done. It is the removal of the fraction of tree biomass that largely affects the amount of flooded organic matter in the site. The amount of allochthonous DOC or POC definitely influence carbon processing in the reservoir. From the simulations that we did, it is a linear response. More DOC input, more $CO_2$ emissions. When creating a reservoir, it should not influence the water flow and DOC concentration of upstream. To simulate the variance of DOC concentration in the inflow, this requires a landscape hydrological model and a terrestrial ecosystem model with a detailed DOC module. Therefore, the amount of allochthonous carbon input is beyond the scope of our study.*

6. -The model is a 1-DV model and no vertical profile of modeled variables is shown. Such typical figures are missing to evaluate if how processes are well reproduced by the model or if the model gives "only" a good average value for the "bulk" water column. It would be nice to see data from Teodoru et al (2011) (pelagic and benthic respiration, primary production, benthic respiration) and Demarty et al (2011) (vertical profiles) for instance being used for comparison with the model.

   *Author response: Indeed, this is a 1-D model that typically simulate mean conditions like mean water depth and mean amount of flooded soil organic matter. Due to its 1-D limitation, it is not realistic to compare the modeled results (mean depth ranging 5 to 11 m) to the measurements (the vertical profile ranging up to 30m, as shown in Demarty et al., 2011). Because of the spatial heterogeneity of $CO_2$ fluxes in terms of pre-flooded landscape (Teodoru et al., 2011), we actually do not have enough profile data for the pre-forested site.*

   *Primary production and benthic respiration were empirically estimated rather than measured in the previous studies (Teodoru et al., 2011). We prefer not to compare models to models.*

   *Reference:*
   *Demarty M, Bastien J, Tremblay A. Annual follow-up of gross diffusive carbon dioxide and methane emissions from a boreal reservoir and two nearby lakes in Quebec, Canada. Biogeosci. 2011; 8: 41-53.*

   *Teodoru CR, Prairie YT, del Giorgio PA. Spatial heterogeneity of surface CO2 fluxes in a newly created Eastmain-1 reservoir in northern Québec, Canada. Ecosystems 2011; 14: 28-46.*

7. -I would recommended to put the monitoring of the pCO2 in the generation station (supplemental) in the main document since it is the best way to have the average concentration over the whole water column. It also offers the possibility of computing downstream emission.

*Author response*: *Did as suggested and necessary explanation on the discrepancy has been added in the text. Reviewer 3 has the similar comment (R3C18).*

*In this study, we tested the model performance for the reservoir eddy covariance tower site where mature forests dominated prior to the flooding. Spatial heterogeneity in surface $CO_2$ fluxes that was linked to the pre-flood landscape types has been reported for the Eastmain-1 reservoir (Teodoru et al., 2011). Thus, the simulated mean water $pCO_2$ has a systematic offset compared to the measured $pCO_2$ in the generation station – i.e. they represent different water.*

*Reference:*
*Teodoru CR, Prairie YT, del Giorgio PA. Spatial heterogeneity of surface CO2 fluxes in a newly created Eastmain-1 reservoir in northern Québec, Canada. Ecosystems 2011; 14: 28-46.*

8.  -A discussion about the pool of carbon fueling emissions would be very interesting: What are the relative contributions of the pelagic respiration, the autochthonous and allochthonous organic matter and the flooded organic matter to the CO2 emissions? Those elements could reinforce the section 4.1 where all sources are listed but no information is given about the main source for the first years and after a few decades.

    *Author response*: *It would be very interesting to investigating their contributions to $CO_2$ emissions. However, this study aims to investigating the flooding effects on post-flooded reservoir surface $CO_2$ emissions. The model calculates the carbon fluxes across the sediment–water interface, which contributes to $CO_2$ emissions by direct DIC fluxes and indirect DOC fluxes.*

    *Unfortunately, the model does not separate the autochthonous and allochthonoud sources of $CO_2$ in the water column, as each modeled water layer has one pool for DIC and DOC, respectively.*

    *We also think that the Figure 5 could show the relative contribution of sediment carbon sources (including flooded terrestrial organic carbon and settled organic carbon from the water column) to total $CO_2$ emissions over time.*

9.  -the section 4.2 is basically about the sensitivity of the model to temperature change on CO2 emissions. I would be very informative to provide illustrations of temperature change on both the physics (vertical stratification, duration of ice cover: : :) and on biogeochemical processes (respiration, PP in the water column, CO2 production in the soils and overlying sediments: : :). Currently, this section does not provide any quantitative.

    *Author response*: *The sensitivity analyses has been done for two parameters (Rw and fo2) and two climate variables (air temperature and wind speed). We have listed the reason why we chose these four parameters in revised section 2.4. Please also see our response to R1 C3 and R3C4 (R: Reviewer, C: Comment).*

*The effects of air temperature and wind speed on thermal dynamics (like vertical temperature, duration of ice cover) have been reported by Wang et al., 2016. Because of the scope (the effects of flooding on reservoir $CO_2$ emissions) of this study, the sensitivity analysis focuses on $CO_2$ emissions and fluxes across sediment–water interface (Figure 5). Since we do not have direct measurements in PP, respiration, $CO_2$ production in the sediment, we feel it's not suitable for such an analysis.*

*Reference:*
*Wang W, Roulet NT, Strachan IB, Tremblay A. Modeling surface energy fluxes and thermal dynamics of a seasonally ice-covered hydroelectric reservoir. Sci. Total Environ.550: 793-805, 2016.*

Detailed comments

10. P1L24: ""engineering" reservoir lifetime (100 years)" could be replace vy the widely-used life-time analysis

    *Author response: Did as suggested.*

11. P1-L27: oxygen effects?

    *Author response: We rephrased the phrase. The term of "partitioning coefficient of decomposition production" was used in the revised manuscript.*

12. -P2-L9-10: Many papers by JJ Cole, Carpenter and their teams or the synthesis by Duarte and Prairie (2005) would be more relevant for the prevalence of heterotrophy in aquatic ecosystems.

    *Author response: Did as suggested.*

13. -P2-L11 " water-saturated sediments where the organic matters (e.g., plant biomass, litter, and soil organic matter)": Sediments are different from the flooded organic matter.

    *Author response: We assumed that soils directly became "sediments" once the flooding events occurred.*

14. -P3L14: what are those "minimum inputs" compare to the listed "sophisticated" models?This should be discussed later on in the manuscript.

    *Author response: We now list the difference in inputs in the text.*

15. -P3-L23-26: "Based on limited empirical data, we test the hypothesis that the boreal reservoir will be a net source of CO2 to the 25 atmosphere. We further hypothesize that the exchanges will be the largest in the first one to two decades and will then show little secular change thereafterăĂˇ Ti.e. year-to-year variability around a fairly constant mean" The Eastmain database is not a limited database: 6 years of EC, several field campaigns with floating chamber, DOC, pCO2, respiration, Chloa to cite a few: : And the two hypotheses here are

not hypothesis since those results are well know (Teodoru et al., 2012). The challenge was rather to check if a simple model is able to reproduce the emissions.

*Author response: While the EM-1 data is quite extensive for a boreal reservoir it is a fairly inadequate data set to evaluate a model or reservoir emissions when the lifetime of a reservoir is considered. Further the observational data does not account for the spatial and temporal scales involved EC data provides a high temporal resolution carbon flux data over several $m^2$ of a 600 $km^2$ reservoir, while field campaigns with floating chambers can only provide sporadic information on a very small footprint. So while the observations are probably the most for any boreal reservoir the uncertainties in observation to model comparison are large.*

*The reason why we developed the process-based model is not only to check if a relatively simple model is able to reproduce the emissions but also to enhance our understanding on mechanisms. Although empirical studies (Teodoru et al., 2012) examined similar hypotheses, they lack the ability to examine process level explanations. What we use a process-based model to simulate the carbon cycle change in response to flooding and environmental inputs over 100 years. We got different results and made different conclusions than the empirical studies, hence e think the hypotheses are reasonable. FAQ-DNDC is a structured hypothesis of how we think, based on the literature, a reservoir's C cycles operates.*

*Reference:*

*Teodoru, C. R., Bastien, J., Bonneville, M.-C., del Giorgio, P. A., Demarty, M., Garneau, M., Hélie, J.-F., Pelletier, L., Prairie, Y. T., Roulet, N. T., Strachan, I. B., and Tremblay, A.: The net carbon footprint of a newly created boreal hydroelectric reservoir, Global Biogeochem. Cycles, 26, GB2016, 10.1029/2011GB004187, 2012.*

16. -P4-L21: Is the sentence a title for a section?

    *Author response: We re-wrote the sentence.*

17. -Page 10 Line 20-23: There is no explanation about the tree removal. Was it really done before flooding? If yes, this should be in the site description. Is it a theoretical hypothesis for the evaluation of the role of tree trunk organic matter on emissions and the evaluation of mitigation options?

    *Author response: We added the information about the tree removal in the site description. Some trees were removed in the first winter after the inundation by controlling icepack elevation through dam operations. We think that clear-cutting before flooding would help mitigating $CO_2$ emissions.*

18. -P11-L18: what does dr stands for?

    *Author response: dr is the revised Willmott index. We added the notation when it firstly appears. This was also addressed in response to reviewer 1.*

19. -P11-L26-27: This should be extended as noted is the general comments.

    *Author response: We did as suggested. See our response to major comment 8 above.*

20. -P12-L25-26: "Both increasing and decreasing wind speeds enhanced annual CO2emissions only by 1 and 1% over 100 years, respectively." Unclear sentence, should be rephrased.

    *Author response: We rephrased the sentence. Here we mean "Changing wind speeds by 20% enhanced annual $CO_2$ emission by up to 1% over the simulation period."*

21. -P12-L29: "grater": : : greater

    *Author response: Did as suggested.*

22. -P13-L17: more information about the pelagic processes is needed since this is where the improvement over Kim et al. (2016) are.

    *Author response: We mentioned the FF-DNDC study in our introduction and discussed the difference between two models. Please see our short comments by Weifeng Wang and our response to the comment 2.*

23. -P13-L20: "Our simulations also show that sediment organic C keeps loosing over the simulation period" needs to be rewritten taking into account that this is very probably the pool of flooded organic matter that loose C instead of the sediment which might accumulate C even if at very low rate.

    *Author response: Yes, the continuous carbon loss across the simulation period is attributed to the flooded organic matter. We have revised the sentence to make our conclusion clearer.*

---

## Author Comment (AC3) · 5 Jul 2016

Reviewer #3

We appreciated for the comments and suggestions that significantly improve the quality of the manuscript. We have addressed referee 3's comments point by point and will make changes in the revised manuscript, which are detailed below.

1. The authors represent a modelling approach to quantify CO2 emissions by integrating a terrestrial and an aquatic model. They applied their model framework to assess the effects of reservoir creation and the following CO2 emission on carbon dynamics. In my opinion this work represents an important step towards a more complete understanding of the carbon cycle. It stresses the importance to integrate terrestrial and aquatic systems to fully map the different components of the carbon cycle, which is especially importance with respect to climate change. Although I see several issues that should be solved first, I recommend publishing the manuscript in Biogeosciences after revision.

MAJOR COMMENTS:

2. A) One concern is that to me it seems that the authors did the calibration of their model and the validation not with independent data. It should be made more clear which data have been used for calibration (P1 L16 'using the measurements : : : in a _600 km2 boreal hydroelectric reservoir, Eastmain-1') and for validation ('We then evaluated the model performance against observed CO2 fluxes data from an eddy covariance tower in the middle of the EM-1 reservoir').

   *Author response: We recognize the importance of not evaluating the model on data that was used for calibration, and we did not – we used independent data set to calibrate and validate the model. In our study, we calibrated the model using the mean concentrations of DOC and DIC in the water column and tested the model performance by comparing the outputs against estimates of water column respiration, POC sedimentation, and $CO_2$ fluxes across air-water interface. We do not spilt the $CO_2$ flux data and use part for calibrations because the EC flux data was spotty (only 23% of the measurements from the reservoir were retained – data was rejected because the wind direction was not from the open water surface and/or the conditions were not acceptable such as u\* were too small). We only used observations in our evaluation. To our knowledge, there is no standard gap filling techniques for air-water gas fluxes. Unlike terrestrial ecosystems such as forests and grassland, reservoirs or lakes can store dissolved $CO_2$ and the instantaneous gas exchanges are only controlled not only by metrological factors (e.g., wind) but also the carbon supply (i.e., DIC concentration) from the water column to the surface, and the supply of DIC to the water column. DOC can be easily mineralized. Therefore, we chose these two variables to calibrate the model.*

3. (B) For comparison of observed and simulated data, or rather expected and simulated data I'd like to see more statistical tests to show if the differences are significant or not. Statements as 'reasonably well' (P1, L22) or 'greatly influence' (P12, L28) are not sufficient.

*Author response*: *We did use three statistical index or methods to show the model performance. See P10L29–34. We will add the notation of each statistical index in the text and emphasize them in the abstract. For the statistical analysis, we mentioned in P11L12 that if the mean change over 10% of the base run the model was considered to be sensitive to the parameters tested. This has also been pointed out by reviewer 2.*

4.  (C) The conducted sensitivity analysis only includes Rw, FO2, air temperature and wind speed. It would have been interesting to see how sensitive the model reacts on the most important parameters controlling the processes in the water column, like decomposition rates of POC and DOC.

    *Author response: The sensitivity issue were also raised by the other two reviewers (see our response R1C3 and R2C9, R: reviewer C: comment). The decomposition rates of POC and DOC listed in the table 2 is the decomposition rate at 20 °C, and the real decomposition rates for organic carbon pools in the water column change with temperature. We kept parameter values as same as in their previous model. In our study, we wanted to keep it focus on our major aim, which is the flooding effects on reservoir $CO_2$ emissions.*

5.  (D) It did not appear clear to me, how much of the model developments has been originally done by the authors and how much of their framework relies on the work of others. This should be stated clearly and possibly also indicated in an overview figure as Figure 1.

    *Author response*: *First, we contributed the linkage among the three models. For example, modifications for the water column carbon sub-model listed in section 2.1.1 was to incorporate dynamic thermal and water stratification module.*

    *Second, we modified related components since it is possible to have more dynamic processes through coupling hydrodynamic component and the carbon cycle. Take GPP calculation as an example, the old version of lake carbon model calculated GPP using an empirical function of total phosphorus (constant input value) so there was no dynamics of GPP. We also modified production to include the influence of water temperature (calculated in the thermal module), mixing depth (from the thermal module), chlorophyll-a concentration (newly calculated using a function of POC concentration).*

    *Third, we modify a terrestrial soil biogeochemical model to simulate sediment carbon dynamics (section 2.1.3). Many process have to be added in the new sediment module. Overall, our new model can dynamically simulate reservoir carbon processing and flooding effects, while few models are able to.*

    *We revised the manuscript to make sure that our direct contributions are clear in the overall description of our model.*

6.  (E) I see some potential for improvement in the discussion.- P 13 LL14: The empirical model shows a decline period of 12 to 15 years. Another study (not including water column processes) estimates the period to be several decades. The author's model (including water column processes) estimated a period of only 3 years. It should be more clearly discussed

that this inconsistency (a) shows the importance of including the water column processes and (b) shows that the implementation in the model can still be improved.

*Author response: This is a good suggestion. We, as we responded to comments by the first two reviewers, discussed more about the discrepancy between empirical study (Teodoru et al., 2012), previous modelling study (Kim et al., 2016), and ours.*

*References:*

*Kim, Y., Roulet, N. T., Li, C., Frolking, S., Strachan, I. B., Peng, C., Teodoru, C. R., Prairie, Y. T., and Tremblay, A.: Simulating carbon dioxide exchange in boreal ecosystems flooded by reservoirs, Ecol. Model., 327, 1-17, http://dx.doi.org/10.1016/j.ecolmodel.2016.01.006, 2016.*

*Teodoru, C. R., Bastien, J., Bonneville, M.-C., del Giorgio, P. A., Demarty, M., Garneau, M., Hélie, J.-F., Pelletier, L., Prairie, Y. T., Roulet, N. T., Strachan, I. B., and Tremblay, A.: The net carbon footprint of a newly created boreal hydroelectric reservoir, Global Biogeochem. Cycles, 26, GB2016, 10.1029/2011GB004187, 2012.*

7. The authors state (P13 L25) that they 'did not incorporate methane production'. Please explain why do you think that 'if methane production is included, the reservoir would probably need more time than reported here'.

   *Author response: The development of a methane module requires the incorporation of the cycling of oxygen, the modification of methane production in DNDC, the oxidation of methane in the water column, and the inclusion of convective and ebullition transport. These developments are substantial and beyond the scope of this paper (we have been developing a methane module for several months now and expecting to be publishing this work in the autumn). The statement in the manuscript is an inference: we are speculating that a portion of the methane produced in the sediments will be transported through bubbling. The larger of upward carbon fluxes of DOC, DIC, and $CH_4$, the longer the sediment loses carbon. We revised the paragraph to make the logic and words more precise.*

Minor comments:

8. (A) It would be helpful to show the sequence of processes calculated in the model. I assume this could be added to Figure 1.

   *Author response: Did as suggested.*

9. (B) Some specific questions: Why is GPP depending on DOC (Equ.1)? How did the authors convert form biomass to carbon (P5 LL5)?

   *Author response: The equations 1 and 2 are from Carignan et al. (2000). Their study was conducted for Canadian Shield lakes where our reservoir is located "… a negative effect of DOC on both GP and R. This effect may be due in part to a metabolic inhibition of photosynthesis and respiration by DOC… " is asserted by Carignan et al., 2000.*

*Generally, a coefficient of 0.5 is used for conversion of biomass to carbon in the model. We checked all units for the equations newly added to the model.*

*Reference:*

*Carignan, R., Planas, D., and Vis, C.: Planktonic production and respiration in oligotrophic Shield lakes, Limnol. Oceanogr., 45, 189-199, 10.4319/lo.2000.45.1.0189, 2000.*

Figures:

10. -The figures only partly support the message of the manuscript (e.g. Figure 4 and Figure 5).

    *Author response: We deal with the reviewer's comments below by each figure.*

11. Figure 1 is not clear enough to show the important processes and fluxes in the model. Some arrows seem to come from nowhere (e.g. the solid line arrow 'Incident solar radiation', or one solid black line going down from the 'Passive humus pool'). Please clarify.

    *Author response: We revised the figure 1 to make it clearer.*

12. -In Figure 2 one cannot distinguish the temporal patterns the authors refer to in the text, especially in (c), (d) and (e).

    *Author response: We restructured the figure following the comment (two columns) by referee 1 and changed the symbols (from closed cycle to open triangle) to make it readable.*

13. Figure 3. The model's reaction seems relatively constant over the years, which could be a sign that important processes might be ignored or their implementation should be improved. Please discuss.

    *Author response: The modeled annual emissions are quite close to EC observations and the camber-based observations, with the exception year one chamber fluxes. The trends of the measurements and model output are decreasing over time. However, the standard error in the measurements is large so it is not a highly constrained test except that the simulated fluxes are the same order of magnitude as the observations and they have the same general trend. The large errors in observations is a reality.*

14. Figure 4. I can only partly agree with the conclusions the authors draw from the data shown in this figure, especially for the sinking rate which seems to be poorly reproduced. Why do the observed data don't have any variance?

    *Author response: From the literature (Teodoru et al., 2011), we do not find the variance of reported values. We speculate that the variance of the measurements should be larger than our modeled because the field measurements are typically instantaneous.*

*For the sinking rate, the annual means should not have significant difference, as the simulated mean is within the standard deviation of measured mean. Unfortunately, there is only one year data available.*

*Teodoru, C. R., Prairie, Y. T., and del Giorgio, P. A.: Spatial heterogeneity of surface $CO_2$ fluxes in a newly created Eastmain-1 reservoir in northern Québec, Canada, Ecosystems, 14, 28-46, 10.1007/s10021-010-9393-7, 2011.*

15. Figure 5. The figure does not support the statement that the 'annual CO2 emissions and benthic dissolved CO2 fluxes showed great sensitivity to Rw'. There seems to be some sensitivity but I would not call it 'great'. And besides I'd like to see more statistics on the significance of differences in general.

    *Author response: We defined that the model is sensitive to the tested parameter if the modeled results over 10% of the base run.*

16. Figure 7. What does this figure show? It doesn't fit to the caption.

    *Author response: We have revised the caption. It is "Figure 7: Simulated sediment carbon burial efficiency (= [$F_{DOC}$ + $F_{DIC}$] / $F_{POC}$) under concurrent climate and hydro-thermal regime. If the net flux is from sediments to the overlying water column the burial efficiency is more than 1.".*

17. Figure captions should include more units. E.g. caption for Figure 5: 'The curves were smoothed using a moving average filter with a span of 5.' I assume 5 years. And in the caption for Figure 6 'The curves were smoothed using a moving average filter with a span of 5.' Here I assume days.

    *Author response: Yes, it should be 5 days. We added this information to the caption.*

18. I would rather keep Figure S1 in the main part and remove others (e.g. Figure 2 and Figure 7) to the supplementary.

    *Author response: Reviewer 2 had the similar comment (see R2C7). We completely agree move figure S1 to the main part and add necessary explanation on the discrepancy between modelled and observed. However, figure 2 is to test the model performance for simulating $CO_2$ emissions. Figure 7 is our major finding by using the model to see how long the terrestrial organic carbon is going to influence reservoir carbon cycle.*

(D) Tables:
19. Table 1. 'Thickness of mineral soil' is assumed to be 80cm. Please indicate/explain the basis of this assumption? Especially because it strongly influences the soil organic C in mineral soil and therewith the size of this carbon pool. What are the numbers in parenthesis for 'soil organic C (based on Paré et al 2011)?

*Author response:* *For terrestrial ecosystems, scientists typically assume 1m soil depth. Since we get the organic layer depth (20cm) in our studies area from the literature, we assume that the thickness of mineral soil layer is 80 cm. SOC content in mineral soil is not affected by the thickness.*

20. Table 2: On which data are the assumptions based?

   *Author response:* *We assume no trees will survive in submerged conditions. The fO2 is assumed based on DNDC model calculation for wetlands.*

21. (E) There are some language issues and typos in the manuscript (e.g. caption Figure 6 'celerity' should be 'clarity'; P2 L29 'in a seasonally ice-covered lakes'; P9 L5 'can generates').

   *Author response:* *Thanks. We will double check the typos.*